# MoCL: Metabolic Optimization for Curvature-Aware Continual Learning

**Jiajun Lai** [1]  **Qi Liu** [1]  **Shijie Li** [1]  **Huaiguang Jiang** [1 2 3]

## Abstract

Continual learning requires models to mitigate catastrophic forgetting of prior knowledge while learning a sequence of tasks. Although existing methods based on orthogonal projection prevent interference by constraining parameter updates, they tend to limit plasticity as the task sequence progresses. The reliance on the linear approximation further causes the projected gradients to deviate from the nonlinear manifold. To address these issues, we propose Metabolic Optimization for Continual Learning (MoCL), a rehearsal-free framework that strikes a balance between stability and plasticity. To capture the geometric manifold of prior knowledge, MoCL introduces a factorized subspace approximation that avoids expensive explicit matrix inversion. Given the heavy-tailed distribution of the Fisher Information Matrix, we employ a metabolic gating based on Tsallis entropy to suppress updates that conflict with historical knowledge. Theoretical and empirical analyses show that MoCL suppresses interference while supporting shared low-loss behavior across sequential tasks. Extensive experimental results across multiple benchmarks demonstrate that MoCL outperforms state-of-the-art methods in both classification performance and efficiency.

## 1. Introduction

Current advances in deep neural networks primarily rely on their ability to learn complex patterns from static datasets (Li et al., 2025a; 2026b;a). Through offline training on these independent and identically distributed (i.i.d.) samples (He et al., 2016; Redmon et al., 2016; Vaswani et al., 2017), models can capture the global statistics of a specific task. However, the knowledge acquired in this setting remains strictly confined to fixed data distributions. This limitation prevents models from continually adapting to evolving environments while retaining previously acquired knowledge, which necessitates continual learning (Parisi et al., 2019; van de Ven et al., 2022; Wang et al., 2024; Zhou et al., 2024b). By departing from the i.i.d. assumption, continual learning involves training models on a continuous stream of sequential tasks. This transition introduces the core challenge of catastrophic forgetting (McCloskey & Cohen, 1989; French, 1993; 1999), which refers to the significant degradation in performance of previously learned tasks after adapting to a new task.

Recently, a growing body of research has focused on utilizing pre-trained models, such as Vision Transformers (ViTs) (Dosovitskiy et al., 2021), to mitigate catastrophic forgetting by leveraging their robust generalization capabilities (Wang et al., 2022c; Lu et al., 2024; Wu et al., 2025). Specifically, prevalent approaches employ learnable prompts or low-rank adaptation (LoRA) (Hu et al., 2022) to incorporate new knowledge. Regarding the former, techniques including VPT-NSP$^2$ (Lu et al., 2024) and CPrompt (Gao et al., 2024) introduce specialized optimization constraints or alignment mechanisms to enhance knowledge stability. Alternatively, InfLoRA (Liang & Li, 2024) performs low-rank adaptation for new tasks within an interference-free subspace. Nonetheless, methods based on learnable prompts often suffer from inaccurate task matching in an expanding pool. Conversely, LoRA-based approaches inherently limit model performance due to rigid constraints on update directions. These drawbacks of existing methods prompt the question: **Is there a novel framework that can achieve a favorable balance between stability and plasticity without depending on imprecise retrieval mechanisms or rigid constraints?**

To overcome the aforementioned challenges, prior studies such as EWC (Kirkpatrick et al., 2017) leverage the Fisher information matrix (FIM) to minimize loss in historical tasks, rather than treating the parameter manifold as a flat hyperplane in orthogonal methods based on gradients (Farajtabar et al., 2020; Cheng et al., 2025). However, EWC maintains stability by strictly limiting changes to essential

[1]School of Future Technology, South China University of Technology, Guangzhou, China [2]Research and Development Department, Guangdong Artificial Intelligence and Digital Economy Laboratory (Guangzhou), Guangzhou, China [3]Guangdong Engineering Research Center of Low-Carbon Synthetic Biotechnology, Guangzhou, China. Correspondence to: Huaiguang Jiang <hihuagong2021@scut.edu.cn>.

*Proceedings of the 43$^{rd}$ International Conference on Machine Learning*, Seoul, South Korea. PMLR 306, 2026. Copyright 2026 by the author(s).

parameters, thereby suppressing the plasticity required for new knowledge. In search of a more dynamic equilibrium, we refer to neurobiological findings showing that metabolic regulation is crucial for synaptic homeostasis and lifelong learning (Zenke et al., 2017; Kudithipudi et al., 2022; Liu et al., 2024). Notably, the naked mole-rat employs metabolic suppression to reduce energy expenditure, allowing it to survive under resource constraints (Farhat et al., 2020). Despite its biological significance, the role of metabolic mechanisms in continual learning remains relatively unexplored.

Inspired by Random Matrix Theory (RMT) (Ghorbani et al., 2019; Martin & Mahoney, 2021) and metabolic mechanisms, we propose **M**etabolic **O**ptimization for **C**ontinual **L**earning (MoCL), which employs Tsallis entropy (Tsallis, 1988) to achieve metabolic gating that uses the heavy-tailed curvature structure to suppress sharp sensitive directions while preserving more freedom on sloppy modes. Underpinned by the Kronecker-Factored Approximate Curvature (K-FAC) (Martens & Grosse, 2015) of activations and gradients, MoCL identifies the geometric manifold of consolidated knowledge through an efficient product of compact Kronecker factors plus low-rank momentum and constructs FIM subspaces for each task with greater geometric fidelity than standard first-order approximations. To maintain the stability of the manifold, the metabolic gating evaluates the energy distribution within historical subspaces via nonextensive statistics to compute a modulation factor that attenuates gradient components aligned with sensitive directions inferred from prior knowledge. Consequently, MoCL enables efficient inference without requiring rehearsal, avoiding component selection overhead and the storage of raw samples. Furthermore, it leverages curvature awareness to ensure scalable compatibility with nonlinear layers. This ability effectively overcomes the limitations of previous methods, as summarized in Table 1.

*Table 1.* Comparison of MoCL with state-of-the-art methods on three properties: 1) Rehearsal-free (no raw sample storage), 2) Inference Efficiency (no component selection overhead), and 3) Nonlinear Compatibility (gradient awareness in nonlinear layers).

| Method | Rehearsal-free | Inference Efficiency | Nonlinear Compatibility |
|---|---|---|---|
| L2P (2022c) | ✓ | ✗ | ✗ |
| DualPrompt (2022b) | ✓ | ✗ | ✗ |
| InfLoRA (2024) | ✗ | ✓ | ✗ |
| CoSO (2025) | ✓ | ✓ | ✗ |
| SD-LoRA (2025) | ✓ | ✓ | ✗ |
| **MoCL (Ours)** | ✓ | ✓ | ✓ |

Comprehensive experiments conducted on several standard benchmarks, including ImageNet-R, CIFAR100, DomainNet, and ImageNet-A, confirm the effectiveness of MoCL. The results demonstrate that MoCL outperforms state-of-

the-art methods while requiring low memory and enabling rapid training. Additionally, the combination of high efficiency and robust performance makes MoCL a practical solution for scalable deployment.

In summary, the main contributions of our work include:

- We propose MoCL, which replaces rigid orthogonal projections with elastic metabolic gating guided by second-order curvature information, reducing update interference while preserving plasticity.

- We introduce a factorized subspace approximation to capture second-order curvature while reducing memory overhead and avoiding explicit matrix inversion.

- Experiments on several benchmarks confirm that MoCL outperforms state-of-the-art methods in both accuracy and efficiency.

## 2. Related Work

### 2.1. Continual Learning

Continual learning (Parisi et al., 2019; Masana et al., 2023; Wang et al., 2024) seeks to address catastrophic forgetting of previously learned information, allowing models to acquire new knowledge across a sequence of distinct tasks. Conventionally, methods trained from scratch can be categorized into three main streams: regularization-based (Aljundi et al., 2018; Li & Hoiem, 2018; Lee et al., 2019), expansion-based (Yan et al., 2021; Douillard et al., 2022; Han et al., 2023), and rehearsal-based (Wang et al., 2022a; Ermis et al., 2022; Jeeveswaran et al., 2023). Regularization-based methods assess the importance of each parameter in prior tasks and employ specialized constraints on parameter updates, thereby preventing critical weights from deviating from their optimal values. Expansion-based methods dynamically construct independent model branches to achieve physical isolation between new and old knowledge as the task sequence progresses. Rehearsal-based methods preserve a fraction of historical samples to approximate joint training.

With the development of pre-trained models, their prior knowledge and generalization capabilities have proved effective for continual learning. Based on this, recent studies have focused on parameter-efficient adaptation and optimization strategies. These approaches can be categorized into prompt-based, LoRA, and subspace optimization. Prompt-based methods typically introduce learnable tokens into a frozen backbone. For example, L2P (Wang et al., 2022c) retrieves prompts by matching input features from a shared pool, whereas CPrompt (Gao et al., 2024) targets the alignment of training and testing consistency. Unlike input modulation, LoRA-based methods learn new task weights via low-rank modules. CL-LoRA (He et al., 2025) proposes

a dual-adapter that captures both shared knowledge and unique task features within the sequence separately. Similarly, SD-LoRA (Wu et al., 2025) independently learns both the magnitude and direction of LoRA components. In addition to adapters, GPM (Saha et al., 2021) enforces orthogonal updates with respect to stored gradient subspaces, while SGP (Saha & Roy, 2023) relaxes this rigid constraint through scaled updates along those subspaces. Unlike SGP, MoCL gates updates in factorized Fisher subspaces rather than rescaling first-order gradient subspaces. CoSO (Cheng et al., 2025) explores subspace optimization by conducting orthogonal updates across multiple low-rank subspaces. Beyond these approaches, recent work has also studied continual learning from an optimization and geometry perspective. For example, C-Flat (Bian et al., 2024) and C-Flat Turbo (Li et al., 2025b) improve continual adaptation by encouraging flatter optima, while ZeroFlow (Feng et al., 2025) mitigates forgetting in gradient-free settings. These works offer a complementary view of forgetting and stability, whereas MoCL leverages historical second-order curvature to identify sensitive directions and selectively gate updates, better balancing stability and plasticity.

## 2.2. Low-Rank Training

For a frozen weight matrix $\mathbf{W}_0 \in \mathbb{R}^{d \times k}$, LoRA reparameterizes weight updates as $\Delta \mathbf{W} = \mathbf{B}\mathbf{A}$ by introducing two low-rank matrices $\mathbf{A} \in \mathbb{R}^{r \times k}$ and $\mathbf{B} \in \mathbb{R}^{d \times r}$, enabling low computational and memory cost. However, compared to full fine-tuning, confining weight updates to a low-rank subspace might lead to a decrease in model performance, since the actual $\Delta \mathbf{W}$ is typically not low-rank (Biderman et al., 2024). To address this, recent studies exploit the low-rank structure of gradients or momentum to achieve efficient full-rank training (Zhao et al., 2024; Robert et al., 2025; Shen et al., 2025). For example, CoSO (Cheng et al., 2025) employs GaLore (Zhao et al., 2024) to compress gradients for efficient training. In contrast, MLorc (Shen et al., 2025) directly stores momentum $\mathbf{M}$ in a low-rank format $\tilde{\mathbf{M}}$ via Randomized SVD (RSVD) (Halko et al., 2011), but updates the momentum using full-rank gradients. The combination ensures memory efficiency comparable to Parameter-Efficient Fine-Tuning (PEFT), yet lacks specific mechanisms to mitigate catastrophic forgetting.

## 2.3. Curvature Approximation

Curvature approximation is essential for continual learning because it reveals the key subspace that preserves prior knowledge. However, the computation and storage required to compute the exact curvature can be excessive. To solve this problem, K-FAC (Martens & Grosse, 2015) approximates the FIM as the Kronecker product of the input and gradient covariances. In this context, NCL (Kao et al., 2021) employs K-FAC to approximate the posterior precision ma-

trix for Bayesian weight regularization. Moreover, XK-FAC (Lee et al., 2020) approximates the curvature by considering inter-example correlations. Nevertheless, these methods primarily use curvature information to accelerate convergence, which typically entails high computational and storage costs due to dense statistics and matrix inversions. In contrast, MoCL treats the FIM as a sensitivity map by directly extracting the principal eigenspaces of the input and gradient covariances to construct low-rank factors. This approach focuses on the most significant curvature directions while avoiding explicit inversion and reducing memory usage.

## 3. Methodology

In this section, we first introduce the relevant preliminaries regarding class-incremental learning and Hessian approximation. Subsequently, we detail the proposed MoCL, which is supported by theoretical analysis and empirical validation. The overall framework is summarized in Fig. 1.

### 3.1. Preliminaries

**Class-Incremental Learning.** Class-incremental learning (CIL) is a significant branch of continual learning, characterized by a setting that closely resembles the real world. Specifically, CIL requires a model to learn from a sequence of tasks with different distributions. Consider a sequence of $N$ tasks $\mathcal{T} = \{\mathcal{T}_1, \mathcal{T}_2, \ldots, \mathcal{T}_N\}$, where each task $\mathcal{T}_t$ is associated with a dataset $\mathcal{D}_t = \{(x_i^{(t)}, y_i^{(t)})\}_{i=1}^{|\mathcal{D}_t|}$. Each sample pair $(x_i^{(t)}, y_i^{(t)})$ consists of an input drawn from the input space $\mathcal{X}$ and a label from the label space $\mathcal{Y}_t$. According to the CIL setting, the label spaces of different tasks are mutually disjoint, i.e., $\mathcal{Y}_i \cap \mathcal{Y}_j = \emptyset$ for any $i \neq j$. In this study, we employ a pre-trained ViT (Dosovitskiy et al., 2021) to perform inference in a unified output space $\bigcup_{k=1}^t \mathcal{Y}_k$ without task identities.

**Approximating the Hessian.** In continual learning, quantifying the importance of parameters theoretically requires the Hessian matrix $\mathbf{H}$ of the loss function. However, $\mathbf{H}$ is often indefinite in non-convex optimization and is prohibitively expensive to compute for deep networks. Following prior work, we approximate $\mathbf{H}$ using the FIM (Ritter et al., 2018; Martens, 2020), which measures how sensitive the model likelihood $p(y|\mathbf{x}; \boldsymbol{\theta})$ is to joint changes in the parameters. For a parameter vector $\boldsymbol{\theta} \in \mathbb{R}^D$, the empirical FIM $\mathbf{F} \in \mathbb{R}^{D \times D}$ is defined as follows:

$$\mathbf{F} = \mathbb{E}_{(\mathbf{x},y) \sim \mathcal{D}_t} \left[ \nabla_{\boldsymbol{\theta}} \log p(y|\mathbf{x}; \boldsymbol{\theta}) \nabla_{\boldsymbol{\theta}} \log p(y|\mathbf{x}; \boldsymbol{\theta})^\top \right]$$

To further improve efficiency, existing methods typically employ K-FAC to approximate the FIM as a block-diagonal matrix (Lee et al., 2020; Kao et al., 2021), where each block corresponds to a layer. Apropos of layer $l$, the Fisher block $\mathbf{F}_l$ can be approximated as the Kronecker product of

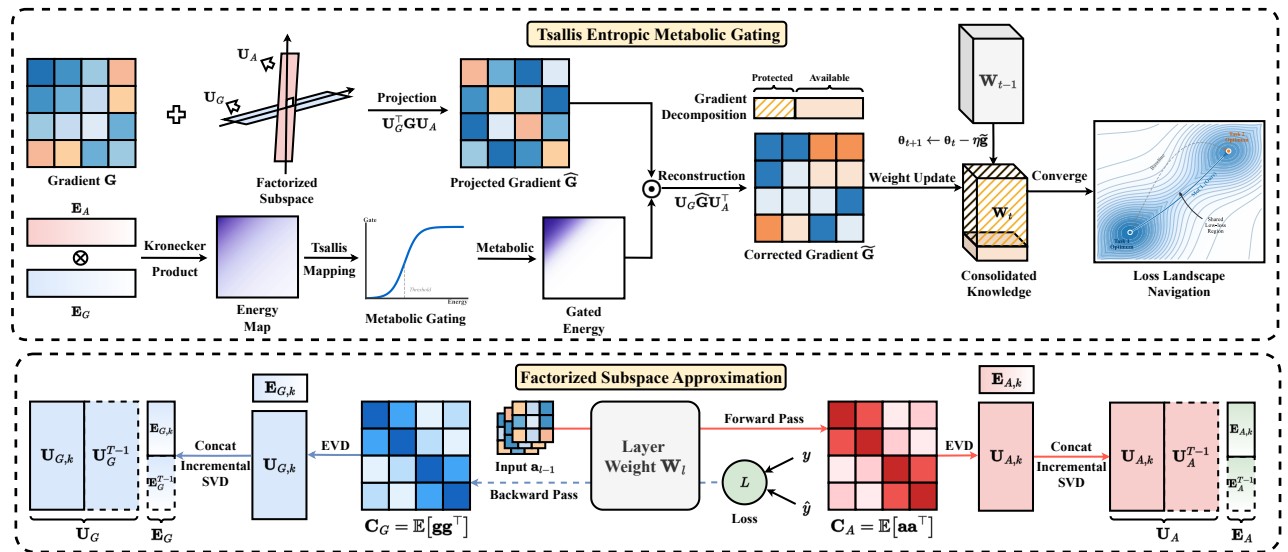

*Figure 1.* Overview of MoCL. To mitigate forgetting, MoCL employs the Tsallis entropy metabolic gating $\mathcal{G}_q$ to modulate gradient updates. During the training, curvature geometry is efficiently approximated via factorized subspaces. These task-specific eigenspaces are subsequently consolidated into the historical knowledge base using Incremental SVD.

the covariance matrix $\mathbf{A}$ of input activations $\mathbf{a}_{l-1}$ and the covariance matrix $\mathbf{G}$ of pre-activation gradients $\mathbf{g}_l$:

$$\mathbf{F}_l \approx \mathbf{A}_{l-1} \otimes \mathbf{G}_l = \mathbb{E}[\mathbf{a}_{l-1}\mathbf{a}_{l-1}^\top] \otimes \mathbb{E}[\mathbf{g}_l\mathbf{g}_l^\top]$$

This approximation preserves parameter correlations while reducing cost compared to the full FIM.

### 3.2. Metabolic Optimization for Continual Learning

From a geometric perspective, catastrophic forgetting is governed by the local curvature of the loss surface around the local optimum of a previous task $\mathcal{T}_{old}$. Specifically, let $\boldsymbol{\theta}^* \in \arg\min_{\boldsymbol{\theta}} \mathcal{L}(\boldsymbol{\theta}, \mathcal{D}_{old})$ denote the optimal parameter obtained after training on $\mathcal{T}_{old}$. When the model subsequently trains on a new task $\mathcal{T}_{new}$, the parameters shift to a new configuration $\boldsymbol{\theta}_{new} = \boldsymbol{\theta}^* + \Delta\boldsymbol{\theta}$. The resulting degradation in performance on $\mathcal{T}_{old}$ can be approximated by a second-order Taylor expansion of the loss function $\mathcal{L}_{old}(\boldsymbol{\theta}) := \mathcal{L}(\boldsymbol{\theta}, \mathcal{D}_{old})$ around the stationary point $\boldsymbol{\theta}^*$:

$$\mathcal{L}_{old}(\boldsymbol{\theta}_{new}) \approx \mathcal{L}_{old}(\boldsymbol{\theta}^*) + \nabla\mathcal{L}_{old}(\boldsymbol{\theta}^*)^\top\Delta\boldsymbol{\theta} + \frac{1}{2}\Delta\boldsymbol{\theta}^\top\mathbf{H}\Delta\boldsymbol{\theta}$$

where $\nabla\mathcal{L}(\boldsymbol{\theta}^*) \approx 0$ since $\boldsymbol{\theta}^*$ is a local optimum. Consequently, the loss increment that characterizes forgetting is determined by the second-order term involving the Hessian matrix $\mathbf{H}$:

$$\mathcal{L}_{old}(\boldsymbol{\theta}_{new}) - \mathcal{L}_{old}(\boldsymbol{\theta}^*) \approx \frac{1}{2}\Delta\boldsymbol{\theta}^\top\mathbf{H}\Delta\boldsymbol{\theta}$$

According to the definitions in Sec. 3.1, $\mathbf{H}$ can be approximated by the empirical FIM $\mathbf{F}$. Therefore, minimizing forgetting becomes equivalent to minimizing the Riemannian metric $\mathcal{R}$ on the parameter manifold (Amari, 1998):

$$\min_{\Delta\boldsymbol{\theta}} \mathcal{R}(\Delta\boldsymbol{\theta}) = \Delta\boldsymbol{\theta}^\top\mathbf{F}\Delta\boldsymbol{\theta}$$

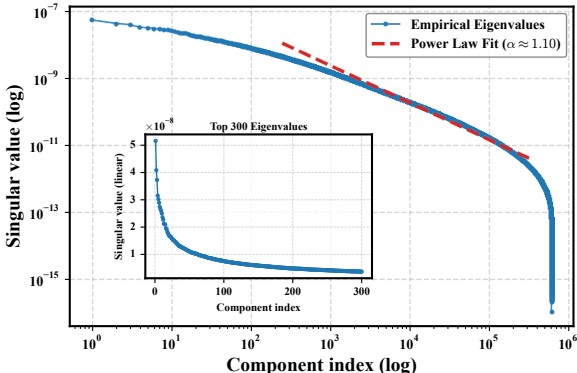

*Figure 2.* Eigenspectrum Analysis. The FIM exhibits a heavy-tailed power-law distribution (red line), decaying from dominant spikes to sloppy modes. This blurred boundary challenges rigid thresholding, motivating our continuous metabolic gating.

By performing the eigendecomposition $\mathbf{F} = \sum_i \lambda_i \mathbf{v}_i \mathbf{v}_i^\top$, where $\lambda$ represents the eigenvalue and $\mathbf{v}$ represents the eigenvector, we can reformulate this objective as:

$$\min_{\Delta\boldsymbol{\theta}} \mathcal{R}(\Delta\boldsymbol{\theta}) = \sum_i \lambda_i(\mathbf{v}_i^\top\Delta\boldsymbol{\theta})^2$$

As the formula revealed, the cost of forgetting is proportional to the magnitude of the eigenvalues. Parameter shifts in directions corresponding to large eigenvalues lead to a sharp increase in $\mathcal{R}$, resulting in catastrophic forgetting. Conversely, directions corresponding to small eigenvalues have little effect on task performance and may facilitate shared low-loss behavior across tasks.

Furthermore, guided by RMT (Ghorbani et al., 2019; Martin & Mahoney, 2021), we observe that the eigenspectrum of $\mathbf{F}$

in deep neural networks does not follow a uniform or compact distribution, but exhibits a heavy-tailed distribution. As shown in Fig. 2, the spectrum decomposes into two distinct regimes: a small number of dominant spikes and a vast bulk of sloppy modes. This continuous power-law decay blurs the topological boundary between signal and noise, rendering fixed thresholding ineffective. As a consequence, traditional orthogonal strategies that treat all eigenvectors as dominant spikes often impose excessive restrictions, thereby limiting plasticity. This restriction prevents the optimization from converging to shared low-loss regions, ultimately trapping the model in a suboptimal solution.

Motivated by this geometric insight, we propose MoCL, a framework inspired by the metabolic efficiency of the naked mole-rat. We map the metabolic suppression mechanism to a gating function $\mathcal{G}_q$ that dynamically adjusts the gradient energy based on the eigenvalues of the FIM. Specifically, we formalize this metabolic suppression as a continuous probability distribution derived from Tsallis statistics, rather than a binary mask used by orthogonal methods. The formulation is represented as follows:

$$\tilde{\mathbf{g}} = \sum_i \mathcal{G}_q(\lambda_i) \cdot (\mathbf{v}_i^\top \mathbf{g}) \mathbf{v}_i \qquad (1)$$

where $\mathbf{g}$ and $\tilde{\mathbf{g}}$ denote the gradient before and after correction, respectively. $\mathcal{G}_q(\lambda_i) \in [0, 1]$ represents the gating coefficient. To provide an interpretable analysis, we derive an upper bound on catastrophic forgetting under MoCL in Prop. 3.1 as stated below.

**Proposition 3.1.** *Assume that the loss function $\mathcal{L}_{old}$ is locally quadratic around the optimal parameters $\boldsymbol{\theta}^*$. Let $\eta$ be the learning rate, and let $\tilde{\mathbf{g}}$ be the modulated gradient defined above. The expected increase in loss on the previous task is bounded by the spectrally weighted energy of the resulting parameter updates:*

$$\mathbb{E}[\Delta\mathcal{L}_{old}] \leq \frac{\eta^2}{2} \sum_i \lambda_i \mathcal{G}_q(\lambda_i)^2 \mathbb{E}[(\mathbf{v}_i^\top \mathbf{g})^2] + \mathcal{O}(\eta^3)$$

**Remark.** The bound reveals that forgetting is resolved by $\lambda_i \mathcal{G}_q(\lambda_i)^2$. For dominant spikes where $\lambda_i$ is large, the gating ensures $\mathcal{G}_q(\lambda_i) \to 0$, which suppresses the term $\lambda_i \mathcal{G}_q^2$ and limits the loss increment. In contrast, for sloppy modes where $\lambda_i \approx 0$, the penalty is negligible regardless of the gating value, allowing $\mathcal{G}_q \to 1$ to preserve plasticity. The detailed proof is provided in Appendix A.

### 3.3. Tsallis Entropic Metabolic Gating

For heavy-tailed Fisher spectra, standard exponential or sigmoid weighting may still assign noticeable weights to dominant curvature directions, which can weaken interference suppression.

**Gate Selection Rationale.** According to Prop. 3.1, forgetting is governed by the term $\lambda_i G(\lambda_i)^2$. A suitable gate should therefore preserve low-curvature directions, suppress dominant spikes, and adapt its shape to the estimated tail of the Fisher spectrum. Tsallis gating satisfies these requirements through a continuous form with finite support when $q < 1$, whereas conventional exponential or sigmoid gates either lack an explicit tail-adaptive parameter or leave nonzero residual weights on sharp directions.

**Metabolic Gating Formulation.** To suppress gradients in high-curvature directions while ensuring plasticity in low-curvature regions, we introduce a metabolic suppression gating based on Tsallis entropy tailored to the heavy-tailed distribution of the FIM spectrum. Firstly, we quantify the tail heaviness of the FIM using the Hill estimator from the Extreme Value Theory (EVT) (Fisher & Tippett, 1928; Hill, 1975). Assuming that the largest $k$ eigenvalues $\{\lambda_1, \lambda_2, \ldots, \lambda_k\}$ determined through spectral knee detection follow a Pareto-type distribution $P(\lambda > x) \propto x^{-1/\xi}$, the Hill estimator $\xi$ is given by:

$$\xi = \frac{1}{k} \sum_{i=1}^{k} (\ln \lambda_i - \ln \lambda_k) \qquad (2)$$

where a larger $\xi$ presents a heavier tail in the spectrum, indicating stricter metabolic suppression to protect crucial knowledge. Based on this, we further recast the Hill estimator in terms of the entropic parameter $q = 1/(1 + \xi)$, which quantifies tail heaviness. In contrast to conventional gating mechanisms that rely on Shannon-Boltzmann entropy (Shannon, 1948), we utilize Tsallis non-extensive statistics to avoid generating probability distributions with infinite exponential tails, ensuring complete interference isolation for highly sensitive parameters. Following the detailed proof in Appendix J, the metabolic gate $\mathcal{G}_q$ can be derived as:

$$\mathcal{G}_q(\lambda) = \left[ 1 - \frac{1-q}{\tau} \cdot \frac{\lambda}{\lambda_{\max}} \right]_+^{\frac{1}{1-q}} \qquad (3)$$

Here, $[\cdot]_+$ denotes the rectification operator and $\tau$ is the temperature scalar. For $q < 1$, Eq. (3) yields a continuous gate that preserves low-curvature modes, suppresses sharp spikes through a finite cutoff, and adapts to the estimated tail heaviness via Eq. (2).

**Empirical Analysis.** Although we quantify the effectiveness of metabolic gating in the ablation study (see Sec. 4.2), it remains to be verified whether metabolic gating indeed alters the optimization geometry as hypothesized. Namely, we hypothesize that the model can converge to shared low-loss regions by leveraging the sloppy modes of the FIM. To validate the hypothesis, we empirically analyze the Linear Mode Connectivity (LMC) between the learned solutions (Frankle et al., 2020; Wu et al., 2025). Specifically, we linearly interpolate between the parameters $\boldsymbol{\theta}_1$ and $\boldsymbol{\theta}_2$ learned

from $\mathcal{T}_1$ and $\mathcal{T}_2$, respectively. The interpolated weights are defined as $\boldsymbol{\theta}_\alpha = (1-\alpha)\boldsymbol{\theta}_1 + \alpha\boldsymbol{\theta}_2$ for $\alpha \in [0,1]$. As shown in Fig. 3, the standard Adam optimizer displays a sharp spike in the loss of $\mathcal{T}_1$ along the interpolation path, indicating that $\boldsymbol{\theta}_2$ has converged to a local minimum that is isolated from $\boldsymbol{\theta}_1$ by a high loss barrier. In contrast, MoCL maintains the loss of $\mathcal{T}_1$ at a consistently low level throughout the interpolation. This empirical observation suggests a lower-loss interpolation path between task solutions, thus preserving knowledge from the previous task.

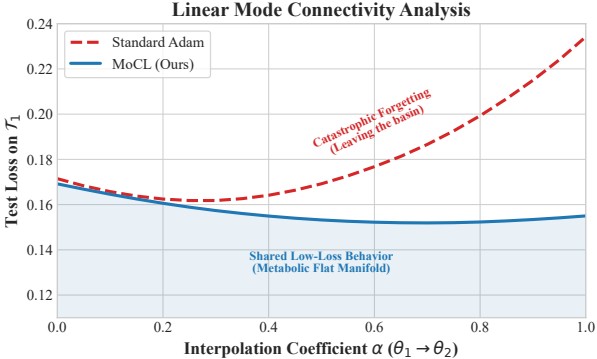

*Figure 3.* We visualize the loss landscape of $\mathcal{T}_1$ as the model parameters are interpolated from $\boldsymbol{\theta}_1$ to $\boldsymbol{\theta}_2$. While standard Adam (Red) suffers from a sharp increase in loss, MoCL (Blue) maintains a consistently low loss.

### 3.4. Factorized Subspace Approximation

**Factorized Gradient Projection.** As discussed in Sec. 3.1, the computational and storage costs of the full FIM are too high to instantiate. Although K-FAC provides a tractable approximation, existing methods use it mainly to accelerate convergence by modifying the gradient space (Kao et al., 2021; Yadav et al., 2025). In contrast, MoCL treats the FIM as a sensitivity map to identify crucial knowledge. To implement this efficiently, MoCL decouples the parameter space into activation and gradient subspaces according to K-FAC, spanned by the eigenvectors $\mathbf{U}_A$ and $\mathbf{U}_G$ with the eigenvalues $\boldsymbol{\lambda}_A$ and $\boldsymbol{\lambda}_G$. By constructing a joint energy map $\mathbf{E} = \boldsymbol{\lambda}_G \otimes \boldsymbol{\lambda}_A^\top$, MoCL quantifies the sensitivity of each direction. Specifically, based on Eq. (1), we project the gradient onto the principal subspace and suppress updates along sensitive directions via the metabolic gating $\mathcal{G}_q$:

$$\tilde{\mathbf{g}} = \mathbf{g} - \mathbf{U}_G \left[ (\mathbf{U}_G^\top \mathbf{g} \mathbf{U}_A) \odot (1 - \mathcal{G}_q(\mathbf{E})) \right] \mathbf{U}_A^\top \quad (4)$$

where $\odot$ denotes the Hadamard product.

**Adaptive Subspace Maintenance.** While K-FAC reduces the complexity of the FIM, there are still two challenges for deep neural networks. (1) Maintaining the full-rank covariance matrices $\mathbf{A}$ and $\mathbf{G}$ increases storage and computational demands. (2) As noted in Sec. 3.2, the FIM spectrum is often heavy-tailed. Retaining the entire eigenspace not only

imposes unnecessary constraints on sloppy modes but also leads to a full-rank subspace as tasks accumulate, thereby hindering the model's plasticity.

To address these challenges, after completing a task, we begin by performing singular value decomposition (SVD) on each covariance factor $\mathbf{C} \in \{\mathbf{A}, \mathbf{G}\}$ to obtain the eigenvalues $\boldsymbol{\lambda}$: $\mathbf{C} = \mathbf{U}\boldsymbol{\Lambda}\mathbf{U}^\top$, where $\boldsymbol{\Lambda} = \mathrm{diag}(\boldsymbol{\lambda})$. Then, we dynamically select the retention rank $r$ based on the cumulative energy threshold $\epsilon$ and an upper bound $r_{max}$:

$$r = \min \left( r_{max}, \ \min \left\{ k : \frac{\sum_{j=1}^k \lambda_j}{\sum_{j=1}^d \lambda_j} \geq \epsilon \right\} \right) \quad (5)$$

where $d$ is the full dimension of the subspace. Based on $r$, we truncate the eigenspace to retain only the top-$r$ principal components, denoted as $\mathbf{U}_r \in \mathbb{R}^{d \times r}$ and $\boldsymbol{\lambda}_r \in \mathbb{R}^r$. This adaptive truncation ensures a compact representation where $r \ll d$. As summarized in Table 2, MoCL significantly reduces the storage complexity from $\mathcal{O}(d^2)$ to $\mathcal{O}(d \cdot r)$ and computational cost from $\mathcal{O}(d^3)$ to $\mathcal{O}(d \cdot r^2)$.

*Table 2.* Comparison of storage and computational complexity for curvature approximation methods.

| Method | Storage | Computation |
|---|---|---|
| Full FIM | $\mathcal{O}(d^4)$ | $\mathcal{O}(d^6)$ |
| Standard K-FAC | $\mathcal{O}(d^2)$ | $\mathcal{O}(d^3)$ |
| **MoCL (Ours)** | $\mathcal{O}(d \cdot r)$ | $\mathcal{O}(d \cdot r^2)$ |

Subsequently, we merge the current eigenspace $\mathbf{U}_r$ into the historical subspace $\mathbf{U}_{old}$. Let $\boldsymbol{\Lambda}_{old}$ and $\boldsymbol{\Lambda}_r$ denote the diagonal eigenvalue matrices corresponding to the historical and current subspaces. The combined subspace $\mathbf{M}_{comb}$, which represents knowledge of all learned tasks, is constructed as:

$$\mathbf{M}_{comb} = \left[ \mathbf{U}_{old}\boldsymbol{\Lambda}_{old}^{1/2}, \quad \mathbf{U}_r\boldsymbol{\Lambda}_r^{1/2} \right] \quad (6)$$

Afterward, we perform SVD and truncation on $\mathbf{M}_{comb}$ to extract the updated basis $\mathbf{U}_{comb}$ and eigenvalues $\boldsymbol{\lambda}_{comb}$. Both $\mathbf{U}_{comb}$ and $\boldsymbol{\lambda}_{comb}$ serve as the updated priors in the gradient projection step shown in Eq. (4). To demonstrate the effectiveness of low-rank approximation, we provide a theoretical analysis in the following Prop. 3.2:

**Proposition 3.2.** *Let $\Delta\boldsymbol{\theta}^\star = -\eta \sum_i \mathcal{G}_q(\lambda_i)(\mathbf{v}_i^\top \mathbf{g})\mathbf{v}_i$ be the parameter update derived from the FIM $\mathbf{F}$ approximated by K-FAC, and let $\Delta\boldsymbol{\theta}_r$ denote the approximate update using the rank-$r$ factorized subspace $\tilde{\mathbf{F}}_r$. Assuming the metabolic gating $\mathcal{G}_q(\lambda)$ is differentiable with a derivative bounded by L, the approximation error is bounded by:*

$$\|\Delta\boldsymbol{\theta}^\star - \Delta\boldsymbol{\theta}_r\|_2 \leq \eta L \lambda_{r+1}(\mathbf{F}) \|P_{\mathcal{S}_{tail}}\mathbf{g}\|_2$$

*where $P_{\mathcal{S}_{tail}}$ is the orthogonal projection of the gradient $\mathbf{g}$ onto the discarded eigensubspace $\mathcal{S}_{tail}$, which is spanned by $\{\mathbf{v}_{r+1}, \ldots, \mathbf{v}_d\}$. The detailed proof and further derivations are provided in Appendix B.*

*Table 3.* Comparison of results on ImageNet-R with different task lengths ($N$). The best results are highlighted in **bold**.

| Method | ImageNet-R ($N = 5$) | | ImageNet-R ($N = 10$) | | ImageNet-R ($N = 20$) | |
|---|---|---|---|---|---|---|
| | $\mathcal{A}_{last} \uparrow$ | $\mathcal{A}_{avg} \uparrow$ | $\mathcal{A}_{last} \uparrow$ | $\mathcal{A}_{avg} \uparrow$ | $\mathcal{A}_{last} \uparrow$ | $\mathcal{A}_{avg} \uparrow$ |
| Joint | $83.58_{(0.04)}$ | / | $83.58_{(0.04)}$ | / | $83.58_{(0.04)}$ | / |
| Full Fine-Tunning | $62.57_{(0.76)}$ | $76.58_{(0.62)}$ | $56.52_{(0.95)}$ | $71.39_{(0.88)}$ | $52.67_{(1.17)}$ | $68.46_{(0.82)}$ |
| L2P | $74.95_{(0.54)}$ | $78.89_{(0.41)}$ | $72.33_{(0.32)}$ | $77.57_{(0.28)}$ | $69.38_{(0.34)}$ | $75.02_{(0.37)}$ |
| DualPrompt | $70.28_{(0.29)}$ | $75.38_{(0.27)}$ | $67.85_{(0.17)}$ | $74.09_{(0.24)}$ | $65.78_{(0.28)}$ | $72.33_{(0.26)}$ |
| CODA-Prompt | $74.52_{(0.21)}$ | $78.65_{(0.25)}$ | $73.11_{(0.31)}$ | $77.25_{(0.26)}$ | $69.63_{(0.24)}$ | $74.03_{(0.29)}$ |
| InfLoRA | $77.25_{(0.18)}$ | $81.29_{(0.14)}$ | $74.55_{(0.37)}$ | $79.12_{(0.36)}$ | $70.22_{(0.33)}$ | $76.62_{(0.34)}$ |
| EASE | $78.08_{(0.18)}$ | $82.63_{(0.25)}$ | $77.45_{(0.20)}$ | $83.05_{(0.23)}$ | $75.53_{(0.25)}$ | $82.66_{(0.24)}$ |
| SD-LoRA | $79.67_{(0.19)}$ | $84.49_{(0.26)}$ | $77.80_{(0.24)}$ | $83.68_{(0.26)}$ | $76.85_{(0.27)}$ | $82.78_{(0.34)}$ |
| CoSO | $81.53_{(0.09)}$ | $84.98_{(0.11)}$ | $80.33_{(0.16)}$ | $84.78_{(0.18)}$ | $77.57_{(0.20)}$ | $82.65_{(0.13)}$ |
| **MoCL (Ours)** | $\mathbf{82.29}_{(0.11)}$ | $\mathbf{86.32}_{(0.09)}$ | $\mathbf{81.33}_{(0.18)}$ | $\mathbf{86.07}_{(0.19)}$ | $\mathbf{79.22}_{(0.17)}$ | $\mathbf{85.03}_{(0.20)}$ |

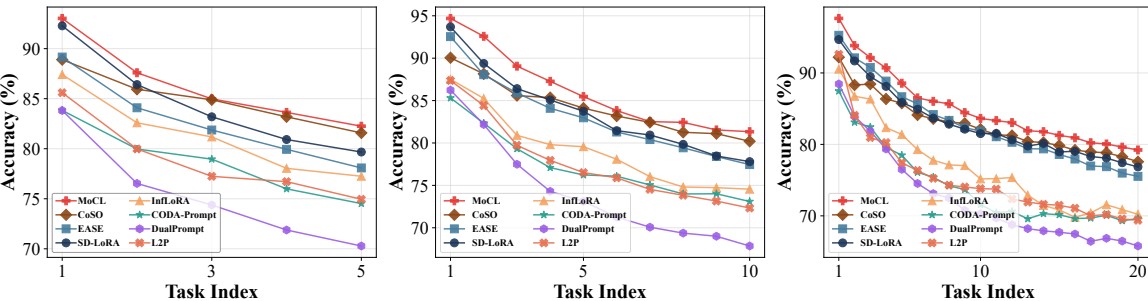

*Figure 4.* The evolution of accuracy on ImageNet-R with different task numbers. (Left): 5 tasks. (Middle): 10 tasks. (Right): 20 tasks.

# 4. Experiments

In this section, we first present the experimental details, followed by a comprehensive set of results across multiple CIL benchmarks.

## 4.1. Experimental Details

**Benchmarks and Evaluation Metrics.** We build on previous studies to assess the performance of MoCL on four datasets (Liang & Li, 2024; Wu et al., 2025), including ImageNet-R (Boschini et al., 2022), ImageNet-A (Hendrycks et al., 2021), CIFAR100 (Krizhevsky et al., 2009), and DomainNet (Peng et al., 2019). The detailed preprocessing procedures for our benchmark datasets are provided in Appendix C.

In order to validate our results, we employ two metrics following the existing methods (Liang & Li, 2024; Wu et al., 2025; Cheng et al., 2025): final accuracy $\mathcal{A}_{last}$ and average accuracy $\mathcal{A}_{avg}$. Specifically, $\mathcal{A}_{last}$ represents the overall performance across all learned tasks after the final task. $\mathcal{A}_{avg}$ denotes the average incremental accuracy during the training process, defined as: $\mathcal{A}_{avg} = \frac{1}{N} \sum_{i=1}^{N} \mathcal{A}_i$ where $\mathcal{A}_i$ is the mean accuracy across all learned tasks after the $i$-th training session.

**Baselines.** To demonstrate the effectiveness of MoCL, we select several state-of-the-art CIL methods for comparison: L2P (Wang et al., 2022c), DualPrompt (Wang et al., 2022b), CODA-Prompt (Smith et al., 2023), InfLoRA (Liang & Li, 2024), EASE (Zhou et al., 2024a), SD-LoRA (Wu et al., 2025), and CoSO (Cheng et al., 2025). To ensure fair comparison, we adopt identical pre-trained models and standardized experimental settings.

**Implementation Details.** In accordance with the established protocol (Wu et al., 2025), we use ViT-B/16 (Dosovitskiy et al., 2021), pre-trained on ImageNet-21K and fine-tuned on ImageNet-1K, as the backbone. During training, only the attention layers are updated using identical Adam optimization (Kingma & Ba, 2015) to ensure fair comparison. More experiment settings are detailed in Appendix D.

## 4.2. Experimental Results

**Comparative Results.** We evaluate the effectiveness of MoCL from two perspectives: performance under different task partitions and comparison across diverse datasets.

As summarized in Table 3, we conduct comprehensive experiments on ImageNet-R with the total number of tasks $N \in \{5, 10, 20\}$. MoCL consistently achieves the best per-

*Table 4.* Comparison of results across different benchmarks. The best results are highlighted in **bold**.

| Method | CIFAR100 ($N = 10$) | | DomainNet ($N = 5$) | | ImageNet-A ($N = 10$) | |
| --- | --- | --- | --- | --- | --- | --- |
| | $\mathcal{A}_{last} \uparrow$ | $\mathcal{A}_{avg} \uparrow$ | $\mathcal{A}_{last} \uparrow$ | $\mathcal{A}_{avg} \uparrow$ | $\mathcal{A}_{last} \uparrow$ | $\mathcal{A}_{avg} \uparrow$ |
| L2P | $84.43_{(0.28)}$ | $89.24_{(0.22)}$ | $70.12_{(0.28)}$ | $76.06_{(0.34)}$ | $46.19_{(0.74)}$ | $56.37_{(1.13)}$ |
| DualPrompt | $82.29_{(0.26)}$ | $87.35_{(0.17)}$ | $68.33_{(0.37)}$ | $74.21_{(0.42)}$ | $50.07_{(0.65)}$ | $59.94_{(0.88)}$ |
| CODA-Prompt | $86.72_{(0.30)}$ | $90.85_{(0.24)}$ | $71.15_{(0.35)}$ | $78.05_{(0.34)}$ | $52.87_{(0.46)}$ | $64.34_{(0.51)}$ |
| InfLoRA | $86.62_{(0.08)}$ | $91.47_{(0.13)}$ | $72.47_{(0.15)}$ | $79.75_{(0.29)}$ | $50.47_{(0.67)}$ | $64.99_{(0.52)}$ |
| EASE | $86.48_{(0.17)}$ | $91.28_{(0.27)}$ | $66.15_{(0.20)}$ | $72.27_{(0.32)}$ | $57.84_{(0.37)}$ | $67.93_{(0.50)}$ |
| SD-LoRA | $86.94_{(0.36)}$ | $91.79_{(0.25)}$ | $72.56_{(0.23)}$ | $78.60_{(0.15)}$ | $56.07_{(0.48)}$ | $65.12_{(0.98)}$ |
| CoSO | $85.21_{(0.14)}$ | $90.94_{(0.11)}$ | $73.10_{(0.12)}$ | $79.87_{(0.16)}$ | $57.27_{(0.26)}$ | $\mathbf{71.27}_{(0.21)}$ |
| **MoCL (Ours)** | $\mathbf{88.12}_{(0.18)}$ | $\mathbf{92.74}_{(0.14)}$ | $\mathbf{74.23}_{(0.14)}$ | $\mathbf{80.22}_{(0.07)}$ | $\mathbf{58.40}_{(0.32)}$ | $68.36_{(0.34)}$ |

*Table 5.* Ablation study of component contributions on ImageNet-R. We compare MoCL against variants with rigid constraints, alternative gating functions, and partial subspace estimation.

| Method Configuration | Change | $\mathcal{A}_{last}$ | $\mathcal{A}_{avg}$ |
| --- | --- | --- | --- |
| **MoCL (Ours)** | **Full Model** | **79.22** | **85.03** |
| *(a) Analysis of Projection* | | | |
| w/ Hard Projection | Soft → Hard | 77.68 | 84.11 |
| w/ Shannon Entropy | Tsallis → Shannon | 78.43 | 84.57 |
| w/ Sigmoid Gating | Tsallis → Sigmoid | 74.88 | 82.54 |
| *(b) Analysis of Subspace* | | | |
| w/ Gradient Only | K-FAC → $G$ only | 76.85 | 83.74 |
| w/ Activation Only | K-FAC → $A$ only | 75.90 | 82.56 |

*Table 6.* Efficiency analysis on ImageNet-R. We report VRAM and GFLOPs compared to other methods.

| Method | VRAM (G) | GFLOPs |
| --- | --- | --- |
| L2P (Wang et al., 2022c) | 16.10 | 70.24 |
| DualPrompt (Wang et al., 2022b) | 13.38 | 70.24 |
| CODA-Prompt (Smith et al., 2023) | 19.61 | 70.24 |
| InfLoRA (Liang & Li, 2024) | 17.98 | 35.12 |
| EASE (Zhou et al., 2024a) | 12.38 | 176.18 |
| SD-LoRA (Wu et al., 2025) | 20.35 | 35.12 |
| CoSO (Cheng et al., 2025) | 10.83 | 35.12 |
| **MoCL (Ours)** | **10.25** | **35.12** |

formance across all partitions. Importantly, the performance advantage of MoCL becomes more pronounced as $N$ increases. Moreover, we present the evolution of accuracy throughout the continual learning process in Fig. 4. As observed, MoCL exhibits a slower rate of accuracy degradation compared to other methods throughout training. This highlights that MoCL can strike a balance between model plasticity and stability.

Furthermore, we discuss the performance of MoCL across different benchmarks as illustrated in Table 4. The result reveals that MoCL still consistently excels and achieves the best $\mathcal{A}_{last}$ on all benchmarks. Although CoSO attains the highest average accuracy $\mathcal{A}_{avg}$ on ImageNet-A, there is a severe performance imbalance between tasks. Specifically, CoSO achieves approximately 82% accuracy on the initial tasks, but its performance drops sharply to approximately 18% on the final tasks. It suggests that the orthogonality strategy imposes excessive constraints, thereby impairing its ability to learn new knowledge. In contrast, MoCL maintains great performance throughout training.

**Ablation Study.** We validate the individual contribution of metabolic gating and factorized subspace on ImageNet-R ($N = 20$), with results summarized in Table 5. Specifically, we employ different projection strategies and sub-

space structures compared to MoCL. We first investigate the influence of gating mechanisms. The results reveal that employing binary hard projection without gating leads to a performance drop of about 1.54% in $\mathcal{A}_{last}$. This suggests that hard projection blocks updates along curvature directions that could facilitate knowledge sharing between tasks. Furthermore, replacing the Tsallis entropy with the Shannon entropy or Sigmoid gating also results in a decrease in accuracy, demonstrating that non-extensive Tsallis statistics are more suitable for modeling the heavy-tailed FIM.

In addition, we evaluate the impact of different subspace structures, including the gradient and activation subspaces. Specifically, when relying solely on the gradient or activation subspace, $\mathcal{A}_{last}$ decreases by 2.37% and 3.32%, respectively. These results validate that K-FAC captures the historical subspaces that encode the task knowledge.

**Efficiency Analysis.** Beyond classification performance, we also analyze computational efficiency on ImageNet-R ($N = 5$) in terms of two key aspects: peak memory usage (VRAM) during training and GFLOPs. As shown in Table 6, by projecting momentum into low-dimensional subspaces (Shen et al., 2025) and using low-rank subspace optimization, MoCL achieves the lowest training memory footprint and inference complexity among all methods. Crucially, these efficiency improvements are achieved without

compromising classification accuracy, making it a scalable framework for continual learning in reality. Detailed implementation of the momentum compression for Adam optimization (Kingma & Ba, 2015) is provided in Appendix G.

**Nonlinear Compatibility.** To verify the nonlinear compatibility discussed in Sec. 1, we extend our method to full-parameter training. Detailed results are provided in Appendix E. The empirical results demonstrate that MoCL not only successfully adapts to nonlinear layers but also achieves superior performance. In contrast, other methods exhibit varying degrees of performance degradation in this training setting.

## 5. Conclusion

In this work, we introduce MoCL to address the limitations of traditional orthogonal projection and LoRA-based methods. By analyzing the heavy-tailed distribution of the FIM, MoCL employs a metabolic gating to dynamically regulate gradient updates, striking a balance between stability and plasticity to effectively mitigate catastrophic forgetting. Furthermore, we introduce a factorized subspace approximation that circumvents the drawbacks of rigid constraints and full-rank operations while maintaining computational efficiency. Comprehensive experiments across several benchmarks demonstrate the effectiveness of MoCL, validating its ability to overcome the linear constraints of previous work.

Although MoCL performs well empirically, several directions for future research remain worth exploring. First, MoCL depends on the quality of the curvature estimation, and more accurate second-order approximations may further improve its robustness and effectiveness. Second, the current adaptive rank truncation in MoCL is effective in practice, while more flexible strategies such as layer-wise rank allocation may further improve efficiency and flexibility. Finally, our study mainly focuses on the CIL setting, and extending MoCL to more challenging scenarios such as online continual learning remains an important direction for future work.

## Impact Statement

This paper presents work whose goal is to advance the field of Machine Learning. There are many potential societal consequences of our work, none which we feel must be specifically highlighted here.

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

# A. Proof of Proposition 3.1

In this section, we provide the detailed derivation of the upper bound for catastrophic forgetting presented in Prop. 3.1.

## A.1. Setup and Assumptions

Let $\boldsymbol{\theta}^*$ be the optimal parameters for the previous task $\mathcal{T}_{old}$. When the model is training on a new task, the parameters are updated from $\boldsymbol{\theta}^*$ to $\boldsymbol{\theta}_{new} = \boldsymbol{\theta}^* + \Delta\boldsymbol{\theta}$. We make the following standard assumptions:

**Assumption A.1.** The model has converged on the $\mathcal{T}_{old}$, meaning that the gradient at $\boldsymbol{\theta}^*$ is negligible, i.e., $\nabla\mathcal{L}_{old}(\boldsymbol{\theta}^*) \approx \mathbf{0}$.

**Assumption A.2.** The loss function $\mathcal{L}_{old}$ is locally convex and smooth, allowing it to be approximated by a second-order Taylor expansion.

**Assumption A.3.** The Hessian matrix $\mathbf{H}$ is approximated by the FIM $\mathbf{F}$, which is positive semi-definite.

## A.2. Derivation

We expand the loss on the previous task $\mathcal{L}_{old}(\boldsymbol{\theta}^* + \Delta\boldsymbol{\theta})$ around $\boldsymbol{\theta}^*$ using a second-order Taylor expansion:

$$\mathcal{L}_{old}(\boldsymbol{\theta}^* + \Delta\boldsymbol{\theta}) \approx \mathcal{L}_{old}(\boldsymbol{\theta}^*) + \Delta\boldsymbol{\theta}^\top \nabla\mathcal{L}_{old}(\boldsymbol{\theta}^*) + \frac{1}{2}\Delta\boldsymbol{\theta}^\top \mathbf{H}\Delta\boldsymbol{\theta} + \mathcal{O}(\|\Delta\boldsymbol{\theta}\|^3) \tag{7}$$

Since $\nabla\mathcal{L}_{old}(\boldsymbol{\theta}^*) \approx \mathbf{0}$, the increase in loss is mainly determined by the quadratic term:

$$\Delta\mathcal{L}_{old} = \mathcal{L}_{old}(\boldsymbol{\theta}_{new}) - \mathcal{L}_{old}(\boldsymbol{\theta}^*) \approx \frac{1}{2}\Delta\boldsymbol{\theta}^\top \mathbf{F}\Delta\boldsymbol{\theta} \tag{8}$$

In MoCL, the parameter is updated by the modulated gradient $\tilde{\mathbf{g}}$ with a learning rate $\eta$, given by $\Delta\boldsymbol{\theta} = -\eta\tilde{\mathbf{g}}$. Substituting this update rule into the loss approximation yields:

$$\Delta\mathcal{L}_{old} \approx \frac{1}{2}(-\eta\tilde{\mathbf{g}})^\top \mathbf{F}(-\eta\tilde{\mathbf{g}}) = \frac{\eta^2}{2}\tilde{\mathbf{g}}^\top \mathbf{F}\tilde{\mathbf{g}} \tag{9}$$

We further utilize the spectral decomposition of the FIM $\mathbf{F} = \sum_j \lambda_j \mathbf{v}_j \mathbf{v}_j^\top$ and the definition of the modulated gradient in the eigenspace of the FIM $\tilde{\mathbf{g}} = \sum_i \mathcal{G}_q(\lambda_i)(\mathbf{v}_i^\top \mathbf{g})\mathbf{v}_i$, where $g_i = \mathbf{v}_i^\top \mathbf{g}$ denotes the projection of the original gradient onto the $i$-th eigenvector. With these representations, the quadratic form in Eq. (9) can be expanded as:

$$\tilde{\mathbf{g}}^\top \mathbf{F}\tilde{\mathbf{g}} = \left(\sum_i \mathcal{G}_q(\lambda_i)g_i\mathbf{v}_i\right)^\top \left(\sum_j \lambda_j \mathbf{v}_j \mathbf{v}_j^\top\right)\left(\sum_k \mathcal{G}_q(\lambda_k)g_k\mathbf{v}_k\right)$$
$$= \sum_i \sum_k \mathcal{G}_q(\lambda_i)g_i\mathcal{G}_q(\lambda_k)g_k\left(\mathbf{v}_i^\top \mathbf{F}\mathbf{v}_k\right) \tag{10}$$

Since the eigenvectors are orthogonal, specifically $\mathbf{v}_i^\top \mathbf{v}_k = 0$ for $i \neq k$, and satisfy $\mathbf{F}\mathbf{v}_k = \lambda_k\mathbf{v}_k$, the inner term can be expressed as:

$$\mathbf{v}_i^\top \mathbf{F}\mathbf{v}_k = \mathbf{v}_i^\top(\lambda_k\mathbf{v}_k) = \lambda_k(\mathbf{v}_i^\top \mathbf{v}_k) = \begin{cases} \lambda_i & \text{if } i = k \\ 0 & \text{if } i \neq k \end{cases} \tag{11}$$

Consequently, all cross-terms vanish, and the double sum reduces to a single sum over the diagonal elements:

$$\tilde{\mathbf{g}}^\top \mathbf{F}\tilde{\mathbf{g}} = \sum_i \mathcal{G}_q(\lambda_i)^2 g_i^2 \lambda_i \tag{12}$$

Finally, by taking the expectation over the stochastic gradients (Bottou et al., 2018) and incorporating them into the forgetting equation, we can obtain the bound below:

$$\mathbb{E}[\Delta\mathcal{L}_{old}] \approx \frac{\eta^2}{2}\sum_i \lambda_i\mathcal{G}_q(\lambda_i)^2\mathbb{E}[g_i^2] \tag{13}$$

By treating the higher-order terms from the Taylor expansion as $\mathcal{O}(\eta^3)$, we obtain the final bound stated in Proposition 3.1.

**Remark.** For theoretical tractability, we formulate the update rule using the standard Gradient Descent (Ruder, 2016). In our practical implementation of MoCL, we adopt Adam that uses $\tilde{\mathbf{g}}$ as input. Given that metabolic gating and subspace projection are applied directly to the gradient before it enters the momentum state, analyzing the properties of $\tilde{\mathbf{g}}$ is sufficient to validate our method.

# B. Proof of Proposition 3.2

In this section, we provide the detailed proof for the error bound of the adaptive subspace truncation presented in Prop. 3.2. This proof establishes that the gradient deviation is resolved by the smoothness of the metabolic gating and the magnitude of the discarded spectral components.

## B.1. Setup and Notation

To isolate the error introduced by subspace truncation from the approximation error of K-FAC, we define the reference matrix $\mathbf{F} \in \mathbb{R}^{d \times d}$ as the FIM estimated by K-FAC. Let its eigendecomposition be $\mathbf{F} = \sum_{i=1}^{d} \lambda_i \mathbf{v}_i \mathbf{v}_i^\top$, where the eigenvalues are sorted as $\lambda_1 \geq \lambda_2 \geq \cdots \geq \lambda_d \geq 0$.

The ideal update $\Delta\boldsymbol{\theta}^\star$ based on the full-rank K-FAC and the approximate update $\Delta\boldsymbol{\theta}_r$ using the rank-$r$ factorized approximation $\tilde{\mathbf{F}}_r$ are respectively defined as:

$$\Delta\boldsymbol{\theta}^\star = -\eta \sum_{i=1}^{d} \mathcal{G}_q(\lambda_i)(\mathbf{v}_i^\top \mathbf{g})\mathbf{v}_i, \quad \Delta\boldsymbol{\theta}_r = -\eta \sum_{i=1}^{d} \mathcal{G}_q(\tilde{\lambda}_i)(\mathbf{v}_i^\top \mathbf{g})\mathbf{v}_i \tag{14}$$

For $\Delta\boldsymbol{\theta}_r$, $\tilde{\lambda}_i = \lambda_i$ if $i \leq r$ and otherwise $\tilde{\lambda}_i = 0$ if $i > r$.

## B.2. Derivation

We define the update error vector as $\boldsymbol{e} = \Delta\boldsymbol{\theta}^\star - \Delta\boldsymbol{\theta}_r$. Since the components for $i \leq r$ are identical, the error arises solely from the discarded tail indices $i = r+1, \ldots, d$:

$$\boldsymbol{e} = -\eta \sum_{i=r+1}^{d} \left[\mathcal{G}_q(\lambda_i) - \mathcal{G}_q(0)\right] (\mathbf{v}_i^\top \mathbf{g})\mathbf{v}_i \tag{15}$$

Assume that $\sup_\lambda |\mathcal{G}'_q(\lambda)| \leq L$. According to the Mean Value Theorem, for each $\lambda_i$, there exists $\xi_i \in (0, \lambda_i)$ such that:

$$|\mathcal{G}_q(\lambda_i) - \mathcal{G}_q(0)| = |\mathcal{G}'_q(\xi_i)| \cdot \lambda_i \leq L \cdot \lambda_i \tag{16}$$

Consequently, we can bound the squared Euclidean norm $\|\boldsymbol{e}\|_2^2$ as follows:

$$\begin{aligned}
\|\boldsymbol{e}\|_2^2 &= \eta^2 \sum_{i=r+1}^{d} \left(\mathcal{G}_q(\lambda_i) - \mathcal{G}_q(0)\right)^2 (\mathbf{v}_i^\top \mathbf{g})^2 \\
&\leq \eta^2 L^2 \sum_{i=r+1}^{d} \lambda_i^2 (\mathbf{v}_i^\top \mathbf{g})^2
\end{aligned} \tag{17}$$

Given the eigenvalues in descending order, we have $\lambda_i \leq \lambda_{r+1}$ for any $i \geq r+1$. Substituting this upper bound yields:

$$\|\boldsymbol{e}\|_2^2 \leq \eta^2 L^2 \lambda_{r+1}^2 \sum_{i=r+1}^{d} (\mathbf{v}_i^\top \mathbf{g})^2 \tag{18}$$

We recognize the summation term as the squared norm of the gradient projected onto the discarded subspace $\mathcal{S}_{\text{tail}} = \text{span}\{\mathbf{v}_{r+1}, \ldots, \mathbf{v}_d\}$. Let $P_{\mathcal{S}_{\text{tail}}}$ denote the orthogonal projection in this subspace. Consequently, the summation term is given by:

$$\sum_{i=r+1}^{d} (\mathbf{v}_i^\top \mathbf{g})^2 = \|P_{\mathcal{S}_{\text{tail}}} \mathbf{g}\|_2^2 \tag{19}$$

Substituting this into Eq. (18), we can obtain:

$$\|\boldsymbol{e}\|_2^2 \leq \eta^2 L^2 \lambda_{r+1}^2 \|P_{\mathcal{S}_{\text{tail}}} \mathbf{g}\|_2^2 \tag{20}$$

Taking the square root yields the final bound:

$$\|\Delta\boldsymbol{\theta}^\star - \Delta\boldsymbol{\theta}_r\|_2 \leq \eta L \lambda_{r+1} \|P_{\mathcal{S}_{\text{tail}}} \mathbf{g}\|_2 \tag{21}$$

**Remark.** It is important to acknowledge that the total deviation from the true Riemannian gradient involves two distinct sources of error:

$$\text{Total Error} \leq \underbrace{\|\text{True Fisher} - \text{Full K-FAC}\|}_{\text{Estimation Error}} + \underbrace{\|\text{Full K-FAC} - \text{Truncated Subspace}\|}_{\text{Truncation Error (Bounded by Prop. 3.2)}}$$

While the first term depends on the inherent quality of the K-FAC approximation, our proposition rigorously bounds the second term. The use of the projected gradient norm $\|P_{\mathcal{S}_{\text{tail}}} \mathbf{g}\|_2$ further tightens this bound, revealing that the effective truncation error is often negligible in practice due to the concentration of gradient energy in the principal subspace.

## C. Dataset Preprocessing

Consistent with previous work (Liang & Li, 2024; Wu et al., 2025; Cheng et al., 2025), we partition the dataset into $N$ tasks. Specifically, for ImageNet-R, we evaluate three distinct settings with $N \in \{5, 10, 20\}$, each containing 40, 20, and 10 classes per task, respectively. Similarly, we divide CIFAR100 into 10 tasks, each comprising 10 classes. Regarding DomainNet, which consists of 345 classes across six distinct domains, we split it into 5 tasks, each containing 69 classes. For ImageNet-A, we organize its 200 classes into a sequence of $N = 10$ tasks, with 20 distinct classes per task. Additionally, to ensure a fair comparison, we apply identical data augmentations across all methods.

## D. Experimental Setups and Implementation Details

We implement all methods using PyTorch on an NVIDIA A800 GPU with 80GB memory. To ensure reproducibility, we fix the random seed to 1993 and the batch size to 128 for most experiments, though L2P, DualPrompt, and EASE require smaller batch sizes. The results are repeated three times to report the mean and standard deviation. However, to ensure a fair comparison in the efficiency analysis (Sec. 4.2), we standardized the batch size to 128 for all methods. In addition, all methods use the Adam optimizer with hyperparameters strictly following the original papers or specific configurations for the ImageNet-R benchmark. Regarding the rank $k$ in metabolic gating, we employ the maximum deviation method to adaptively select it. We present the detailed hyperparameters of all benchmarks in Table 7, while Appendix H provides a sensitivity analysis of the metabolic temperature $\tau$ and the tail-estimation range $k$.

*Table 7.* Hyperparameter settings for different datasets.

| Hyperparameter | CIFAR100 | ImageNet-R | DomainNet | ImageNet-A |
|---|---|---|---|---|
| Training epoch | 20 | 35 | 5 | 35 |
| Metabolic temperature ($\tau$) | 1e-4 | 1e-4 | 1e-4 | 1e-4 |
| Max rank of FIM ($r_{max}$) | 100 | 120 | 160 | 120 |
| Threshold of FIM ($\epsilon_{th}$) | 0.95 | 0.95 | 0.95 | 0.90 |
| Rank of momentum | 15 | 50 | 70 | 50 |

## E. Validation of Nonlinear Layer Adaptation Capabilities

Existing approaches typically fail to adapt to nonlinear layers. First, methods based on gradient orthogonality assume that the parameter manifold is locally linear, a premise that breaks down when applied to nonlinear layers. Second, prompt-based strategies generally restrict optimization to input-level modulation within a frozen backbone, thereby lacking the ability to support deep parameter evolution. Consequently, methods such as L2P and DualPrompt are marked as N/A in Table 8, as they are incompatible with full-parameter training. Finally, most LoRA-based methods incur excessive memory overhead owing to extra modules, rendering full-parameter training computationally prohibitive for tasks with long sequences. In

*Table 8.* Validation of nonlinear compatibility under full-parameter training setting on ImageNet-R with different task lengths.

| Method | ImageNet-R ($N = 5$) | | ImageNet-R ($N = 10$) | | ImageNet-R ($N = 20$) | |
|---|---|---|---|---|---|---|
| | $\mathcal{A}_{last} \uparrow$ | $\mathcal{A}_{avg} \uparrow$ | $\mathcal{A}_{last} \uparrow$ | $\mathcal{A}_{avg} \uparrow$ | $\mathcal{A}_{last} \uparrow$ | $\mathcal{A}_{avg} \uparrow$ |
| L2P | N/A | N/A | N/A | N/A | N/A | N/A |
| DualPrompt | N/A | N/A | N/A | N/A | N/A | N/A |
| InfLoRA | 75.77 | 81.31 | 73.48 | 79.17 | 70.13 | 76.01 |
| SD-LoRA | 79.84 | 84.56 | OOM | OOM | OOM | OOM |
| CoSO | 78.90 | 84.23 | 73.55 | 81.19 | 72.12 | 78.34 |
| **MoCL (Attn-Only)** | **82.21** | **86.28** | **81.23** | **86.04** | **79.09** | **84.98** |
| **MoCL (Full-Param)** | **83.02** | **87.18** | **81.75** | **86.32** | **79.45** | **85.14** |

contrast, MoCL leverages the FIM to capture the second-order curvature of the loss landscape. Coupled with metabolic gating, MoCL enables geometry-aware modulation. To validate the nonlinear compatibility of MoCL, we further conduct comprehensive experiments under a unified full-parameter training protocol on ImageNet-R.

As shown in Table 8, compared with the result in Table 3, InfLoRA and CoSO exhibit varying degrees of performance degradation in different task lengths. In addition, although SD-LoRA achieves great performance at $N = 5$, it experiences Out-of-Memory (OOM) failures at $N = 10$ and $N = 20$ due to an accumulated memory overhead of approximately 6GB per task. Conversely, MoCL maintains stability and even achieves slightly better results in full-parameter training without extensive tuning, proving its effectiveness in adapting to nonlinear layers.

# F. Algorithm for Fisher Information Matrix Approximation

In this section, we detail the concrete implementation of the FIM approximation. Instead of freezing the backbone and performing a separate calibration phase after training, we adopt an online accumulation strategy to dynamically collect curvature statistics during the final training phase.

---

**Algorithm 1** Online Subspace Calibration via K-FAC Statistics

---

**Input:** Calibration data stream $\mathcal{D}_{cal}$, Model $\mathcal{M}$ with layers $\mathcal{L}$
**Output:** Covariance matrices $\mathbf{A}_l, \mathbf{G}_l$ for each layer $l \in \mathcal{L}$
Initialize $\mathbf{A}_l \leftarrow \mathbf{0}, \mathbf{G}_l \leftarrow \mathbf{0}$ for all $l$
$N \leftarrow 0$
**for** each batch $(\mathbf{x}, \mathbf{y})$ **in** $\mathcal{D}_{cal}$ **do**
  Perform forward pass with $\mathbf{x}$
  **for** each layer $l \in \mathcal{L}$ **do**
    Extract input activations $\mathbf{a}_l$
    Update $\mathbf{A}_l \leftarrow \mathbf{A}_l + \mathbf{a}_l \mathbf{a}_l^\top$
  **end for**
  Compute loss and perform backward pass
  **for** each layer $l \in \mathcal{L}$ **do**
    Extract output gradients $\mathbf{g}_l$
    Update $\mathbf{G}_l \leftarrow \mathbf{G}_l + \mathbf{g}_l \mathbf{g}_l^\top$
  **end for**
  $N \leftarrow N + \text{batch\_size}$
**end for**
**for** each layer $l \in \mathcal{L}$ **do**
  Normalize $\mathbf{A}_l \leftarrow \mathbf{A}_l/N$ and $\mathbf{G}_l \leftarrow \mathbf{G}_l/N$
  Compute Eigendecomposition for $\mathbf{A}_l$ and $\mathbf{G}_l$
**end for**

---

### F.1. Online Accumulation Strategy

Let $N_{total}$ be the total epochs of a task. According to the $N_{total}$, we collect curvature statistics at the end of $N_{cal}$ epochs ($N_{cal} = 3$ for CIFAR100, $N_{cal} = 1$ for DomainNet, and $N_{cal} = 5$ for others). During these epochs, we maintain running sums of the covariance matrices for the activations ($\mathbf{A}$) and gradients ($\mathbf{G}$) for each target layer:

$$\mathbf{A}_{t+1} = \mathbf{A}_t + \mathbf{a}_t \mathbf{a}_t^\top, \quad \mathbf{G}_{t+1} = \mathbf{G}_t + \mathbf{g}_t \mathbf{g}_t^\top \tag{22}$$

where $a_t$ and $g_t$ represent the activation and gradient at step $t$, respectively. After training, we normalize $\mathbf{A}$ and $\mathbf{G}$ by the number of samples, then decompose them to obtain the principal eigenspaces. The overall procedure is summarized in Algorithm 1.

### F.2. Theoretical Justification

The validity of the online accumulation strategy relies on the Quasi-Stationary Assumption (Karakida et al., 2019).

**Convergence to Local Optima.** In the end of training, the model parameters $\theta_t$ have converged to the neighborhood of a local minimum $\theta^*$. At this stage, the magnitude of parameter updates $\|\Delta\theta\| = \|\eta \cdot \nabla\mathcal{L}\|$ is negligible, especially given the learning rate decay typical of this phase.

**Stability of Feature Distributions.** Since $\theta_t \approx \theta_{t+1} \approx \theta^*$ at the end of training, the distribution of activations $P(\mathbf{a}|\mathbf{x}, \theta)$ and gradients $P(\mathbf{g}|\mathbf{x}, \mathbf{y}, \theta)$ remains statistically stationary. Therefore, averaging the statistics over these dynamic steps serves as a Monte Carlo approximation (Kunstner et al., 2019) of the expected Fisher Information at the optimal point $\theta^*$:

$$\mathbb{E}_{\theta \sim \mathcal{N}(\theta^*, \epsilon)}[\mathbf{g}\mathbf{g}^\top] \approx \mathbb{E}_{\mathcal{D}}[\nabla \log p(y|x; \theta^*) \nabla \log p(y|x; \theta^*)^\top] \tag{23}$$

This strategy not only eliminates the computational overhead of additional forward and backward passes but also serves as a smoothing mechanism, reducing the variance in curvature estimation caused by mini-batch stochasticity.

## G. Detailed Implementation of the MLorc

To ensure training efficiency comparable to PEFT methods such as LoRA, we apply the momentum compression technique of MLorc (Shen et al., 2025) based on Adam. Unlike GaLore (Zhao et al., 2024), MLorc compresses the momentum to reduce memory overhead instead of compressing the gradient. Specifically, the standard Adam optimizer needs to store first-order $\mathcal{M} \in \mathbb{R}^{d \times k}$ and second-order $\mathcal{V} \in \mathbb{R}^{d \times k}$ momentum matrices, which occupies twice the memory of the updated parameter. To address this problem, MLorc performs the Randomized SVD (RSVD) to reduce the memory usage (Halko et al., 2011). For a target $r \ll \min(d, k)$, both $\mathcal{M}$ and $\mathcal{V}$ are approximated as $\mathbf{U}\mathbf{S}\mathbf{V}^\top$, where $\mathbf{U} \in \mathbb{R}^{d \times r}$, $\mathbf{S} \in \mathbb{R}^{r \times r}$, $\mathbf{V} \in \mathbb{R}^{k \times r}$. At timestep $t$, we first reconstruct the full-rank momentum approximations from the stored low-rank factors of timestep $t - 1$:

$$\begin{aligned}
\tilde{\mathcal{M}}_{t-1} &= \mathbf{U}_{m,t-1}\mathbf{S}_{m,t-1}\mathbf{V}_{m,t-1}^\top \\
\tilde{\mathcal{V}}_{t-1} &= \mathbf{U}_{v,t-1}\mathbf{S}_{v,t-1}\mathbf{V}_{v,t-1}^\top
\end{aligned} \tag{24}$$

Here, $\tilde{\mathcal{M}}_{t-1}$ and $\tilde{\mathcal{V}}_{t-1}$ represent the first-order and second-order momentum after reconstruction, respectively. Subsequently, we update the momentum according to the standard Adam rule using the full-rank gradient $G_t$:

$$\begin{aligned}
\mathcal{M}_t &= \beta_1 \tilde{\mathcal{M}}_{t-1} + (1 - \beta_1)G_t \\
\mathcal{V}_t &= \beta_2 \tilde{\mathcal{V}}_{t-1} + (1 - \beta_2)G_t^2
\end{aligned} \tag{25}$$

$\mathcal{M}_t$ and $\mathcal{V}_t$ are then used to update the model parameters. Ultimately, we compress $\mathcal{M}_t$ and $\mathcal{V}_t$ into low-rank factors via RSVD to reduce memory overhead for the next iteration:

$$\begin{aligned}
(\mathbf{U}_{m,t}, \mathbf{S}_{m,t}, \mathbf{V}_{m,t}) &\leftarrow \text{RSVD}(\mathcal{M}_t, r) \\
(\mathbf{U}_{v,t}, \mathbf{S}_{v,t}, \mathbf{V}_{v,t}) &\leftarrow \text{RSVD}(\mathcal{V}_t, r)
\end{aligned} \tag{26}$$

## H. Sensitivity to Metabolic Temperature $\tau$

In this section, we focus our sensitivity analysis on the metabolic temperature $\tau$, which determines the sharpness of the metabolic gating. According to Eq. (3), as $\tau \to 0$, the gating approaches a hard binary gating, which can be formalized:

$$\mathcal{G}_q(\lambda) = \begin{cases} 1 & \lambda = 0 \\ 0 & \lambda > 0 \end{cases} \tag{27}$$

As $\tau$ increases, the constraining effect of the gating mechanism gradually decreases, leading the optimization to degenerate into the standard Adam fine-tuning. To evaluate the sensitivity of $\tau$, we vary $\tau$ from $10^{-6}$ to $10^{-2}$ and perform experiments on ImageNet-R ($N = 20$). The evolution of the final accuracy $\mathcal{A}_{last}$ and the average accuracy $\mathcal{A}_{avg}$ is revealed in Fig. 5.

As illustrated in Figure 5, both $\mathcal{A}_{last}$ and $\mathcal{A}_{avg}$ reach their peak around the optimal temperature. This indicates that rigorous protection of the dominant feature directions is essential to prevent the gradual erosion of prior knowledge, while permitting updates in flat directions is vital to ensure plasticity. When $\tau > 10^{-4}$, we observe a decrease in accuracy and variance. This empirical evidence demonstrates that a high metabolic temperature attenuates the protective strength of the metabolic gating, causing the optimization to degenerate into unconstrained fine-tuning. In contrast, when $\tau$ decreases to a lower temperature, such as $10^{-5}$ and $10^{-6}$, we also observe a slight decrease in accuracy and variance. This suggests that excessive protection imposes overly rigid constraints, restricting the optimization trajectory and restricting the optimization trajectory and hindering the emergence of shared low-loss behavior across tasks.

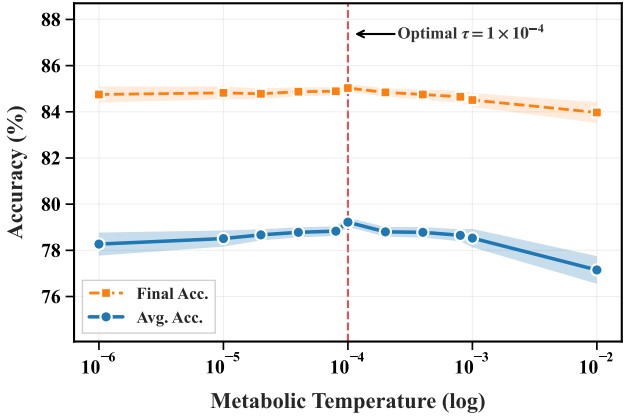

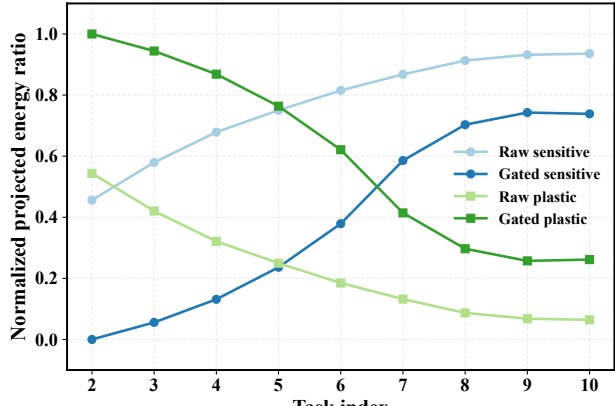

*Figure 5.* Sensitivity analysis of metabolic temperature. Shaded regions indicate the standard deviation over 3 independent runs. The performance peaks at $\tau = 1 \times 10^{-4}$, indicating the optimal trade-off between stability and plasticity.

*Figure 6.* Temporal evolution of gradient flow redistribution across tasks. MoCL suppresses gradient energy on historical sensitive directions while preserving more energy on plastic directions as tasks accumulate.

## I. Temporal Gradient Flow Redistribution across Tasks

Let $\mathbf{U}_{G,t-1}$ and $\mathbf{U}_{A,t-1}$ denote the orthonormal bases of the consolidated historical gradient and activation subspaces before learning task $t$, respectively. To better understand how MoCL redistributes gradient flow across tasks, we measure the historical sensitive component through the factorized core projection

$$\mathbf{C}_t^{\text{raw}} = \mathbf{U}_{G,t-1}^{\top} \mathbf{g}_t \mathbf{U}_{A,t-1}, \qquad \mathbf{C}_t^{\text{gated}} = \mathbf{U}_{G,t-1}^{\top} \tilde{\mathbf{g}}_t \mathbf{U}_{A,t-1} \tag{28}$$

We then define the normalized energy ratios along sensitive directions as

$$R_{\text{sens}}^{\text{raw}}(t) = \frac{\|\mathbf{C}_t^{\text{raw}}\|_F^2}{\|\mathbf{g}_t\|_F^2}, \qquad R_{\text{sens}}^{\text{gated}}(t) = \frac{\|\mathbf{C}_t^{\text{gated}}\|_F^2}{\|\tilde{\mathbf{g}}_t\|_F^2} \tag{29}$$

and define the plastic ratios as their complements:

$$R_{\text{plastic}}^{\text{raw}}(t) = 1 - R_{\text{sens}}^{\text{raw}}(t), \qquad R_{\text{plastic}}^{\text{gated}}(t) = 1 - R_{\text{sens}}^{\text{gated}}(t) \tag{30}$$

As shown in Fig 6 on ImageNet-R ($N = 10$), as tasks accumulate, the gated gradients consistently exhibit less update energy aligned with historical sensitive directions than the raw gradients, while assigning a larger fraction of gradient energy to plastic directions. This suggests that MoCL redirects gradient flow away from sensitive historical components toward directions that better support adaptation to new tasks.

## J. Derivation of Metabolic Gating

**Tsallis Gate Derivation.** Under the adopted variational principle, the Tsallis gate arises as the solution to a well-defined optimization objective, providing a mathematical basis for the gating form in Eq. (3). We interpret metabolic gating $\mathcal{G}_q$ as the allocation of a finite plasticity budget $B$ between curvature modes. Let $p_i \geq 0$ denote the normalized budget allocation with $\sum_i p_i = 1$ and set $g_i = Bp_i$. We obtain the $q$-exponential allocation by maximizing the Tsallis entropy under a $q$-weighted curvature constraint:

$$\max_{\mathbf{p}} \; S_q(\mathbf{p}) \quad \text{s.t.} \; \sum_i p_i = 1, \; \sum_i p_i^q \lambda_i = C, \tag{31}$$

where $S_q(\mathbf{p}) = \frac{\sum_i p_i^q - 1}{1-q}$ and $q < 1$. Applying the KKT conditions yields:

$$p_i \propto [1 - (1-q)\beta(\lambda_i - \lambda_0)]_+^{\frac{1}{1-q}} . \tag{32}$$

where $[\cdot]_+$ enforces the non-negativity $p_i \geq 0$, and $\lambda_0$ accounts for the Lagrange multipliers. Substituting $g_i = Bp_i$ and simplifying the constants can obtain Eq. (3).

**Comparison with Alternative Gates.** To clarify the choice of Tsallis gating, we compare it with two broad types of continuous gates. Gates that remain strictly positive for all $\lambda > 0$ cannot completely remove the contribution of dominant spikes, since $\lambda_i G(\lambda_i)^2 > 0$ still holds on sharp directions. Gates with a fixed shape may provide smooth suppression, but they cannot adapt to different heavy-tail regimes of the Fisher spectrum.

For $q < 1$, the Tsallis gate transitions from values near one for low-curvature to zero beyond a finite threshold. Through Eq. (2), its shape is also linked to the estimated tail of the spectrum. Thus, Tsallis gating combines exact cutoff of sharp spikes with adaptation to the heavy-tailed Fisher spectrum in a single form.

