# OpenReview forum: "MoCL: Metabolic Optimization for Curvature-Aware Continual Learning"
_ICML.cc/2026/Conference — ICML 2026 regular_

### Official Review · Reviewer_Uo5j · 2026-03-06

**Soundness:** 3
**Presentation:** 3
**Significance:** 2
**Originality:** 2
**Overall Recommendation:** 4
**Confidence:** 4

**Summary:**

This paper proposes MoCL, a rehearsal-free continual learning framework that replaces hard orthogonal gradient projection with a soft, Tsallis-entropy-based gating mechanism. It leverages K-FAC to efficiently approximate the FIM's curvature geometry, then applies a continuous gate that suppresses gradient updates in high-curvature directions while permitting updates in flat directions to maintain plasticity. Experiments across four benchmarks show improvements over recent baselines.

**Compliance With Llm Reviewing Policy:**

Affirmed.

**Final Justification:**

I thank the authors for their response. My concerns have been addressed. I am raising my score from 3 to 4. I suggest the revised version include a thorough discussion of the relationship with SGP.

**Key Questions For Authors:**

Q1. The Hill estimator requires selecting k eigenvalues for tail estimation. How is k chosen in practice? How sensitive is the final performance to this choice?

Q2. As subspaces accumulate via incremental SVD, does the effective rank grow to fill the full space, and if so, does the gating degenerate?

**Limitations:**

Yes

**Strengths And Weaknesses:**

Strengths:

1. This paper is well-written and easy to follow.
2. The approach is well-motivated by the empirical observation that the FIM eigen spectrum follows a heavy-tailed power-law distribution. The adoption of Tsallis entropy with compact-support q-exponentials to handle this heavy-tailed structure is theoretically sound.
3. The theoretical framework is relatively complete, with Prop. 3.1 providing an interpretable forgetting bound governed by $\lambda_i$ and Prop. 3.2 separating K-FAC estimation error from truncation error.

Weaknesses:

1. The core motivation of MoCL, that hard orthogonal projection is overly restrictive and should be replaced by continuous importance-weighted gradient scaling, is highly similar to SGP[1]. Both methods recognize that binary projection treats all subspace directions equally, and both propose scaling gradient updates proportionally to eigenvalue-based importance rather than zeroing them out. MoCL differentiates itself through Tsallis entropy and K-FAC, but the high-level insight of soft scaling over hard projection is not new.
2. It seems that some of the results reported for CoSO in this paper may not fully align with those in the original CoSO publication. The authors should clarify whether any differences in experimental settings, data splits, or hyperparameter configurations might account for this, so as to ensure the comparison is conducted on a fully consistent basis.

Reference

[1]. Saha G, Roy K. Continual learning with scaled gradient projection. AAAI 2023.

---

> ### Author Rebuttal · Authors · 2026-03-31
>
> ## Weaknesses
> > **W1:** Overlap with SGP regarding the transition from hard projection to continuous gradient scaling.
>
> **AW1:** We thank the reviewer for noting the connection to SGP. We agree that the relation between MoCL and SGP, together with the closely related prior work like OGD and GPM, should be discussed more explicitly, and we will clarify this positioning in the related work discussion of the revised manuscript.
>
> We agree that both SGP and MoCL are motivated by the limitation of rigid binary projection. However, the soft control is defined differently in the two methods. SGP scales updates along stored subspaces using basis importance derived from representation SVD, whereas MoCL defines soft modulation through Tsallis gating on a factorized curvature subspace, so that update control is guided by sensitivity in the heavy-tailed Fisher spectrum rather than by subspace importance alone.
>
> This distinction is also supported by our ablations. Beyond hard projection, we compare Tsallis gating with Shannon and sigmoid alternatives, and Tsallis performs best. These results suggest that the gain is not solely due to soft scaling, but also to the specific curvature-aware gating design adopted in MoCL.
>
> > **W2:** Some CoSO results differ from the original paper. Clarify differences in experimental settings, data splits, or hyperparameters.
>
> **AW2:** We understand the concern about consistency with CoSO. We clarify that the CoSO results in our paper were reproduced rather than directly taken from the original publication, using the code and configuration released in CoSO’s supplementary material under the same benchmark pipeline as MoCL.
>
> More specifically, CoSO and MoCL were evaluated under the same benchmark setting, including the same data splits, task partitions, backbone initialization, and evaluation protocol. Any minor mismatch with the original CoSO paper may therefore come from reproduction variance, such as random seeds or implementation details, rather than from differences in the benchmark protocol. As noted in the supplementary material, we have provided anonymous source code for transparency.
>
> ---
> ## Questions
> > **Q1:** How is $k$ chosen in practice, and how sensitive is performance to this choice?
>
> **AQ1:** We appreciate the question about $k$. In MoCL, $k$ is not manually tuned for performance. Instead, it is chosen adaptively by detecting the knee point of the Fisher spectrum using the maximum deviation method, which provides a practical rule for choosing how many leading eigenvalues are used to estimate the tail.
>
> To clarify the sensitivity, we perturb the selected $k$ on ImageNet-R ($N=20$) from $0.5k$ to $1.5k$. As shown in Table R1, the performance remains highly stable across this range: $\mathcal{A}\_{last}$ varies only from 78.72 to 79.25, and $\mathcal{A}\_{avg}$ from 84.91 to 85.04. This suggests that MoCL is robust to $k$ within a reasonable range.
>
> **Table R1: Sensitivity to $k$**
> |Metric(%)|$0.5k$|$0.75k$|$0.9k$|$1.0k$|$1.1k$|$1.25k$|$1.5k$|
> |:---|:---:|:---:|:---:|:---:|:---:|:---:|:---:|
> |$\mathcal{A}_{last}$|$78.93_{(0.11)}$|$78.98_{(0.13)}$|$78.72_{(0.22)}$|$79.22_{(0.17)}$|$79.25_{(0.17)}$|$78.93_{(0.16)}$|$79.07_{(0.14)}$|
> |$\mathcal{A}_{avg}$|$84.95_{(0.15)}$|$84.92_{(0.16)}$|$84.95_{(0.20)}$|$85.03_{(0.20)}$|$85.04_{(0.21)}$|$84.91_{(0.18)}$|$84.95_{(0.18)}$|
>
> > **Q2:** Does the effective rank approach the full space as subspaces accumulate, causing gating to degenerate?
>
> **AQ2:** We appreciate the question of whether gating may degenerate as subspaces grow. To address it, we analyze the rank of the consolidated subspace and the projected energy of raw and gated gradients on ImageNet-R over 40 tasks. Here, the raw history rank denotes the accumulated historical span before recompression, and the effective history rank denotes the retained rank after low-rank consolidation, both reported as a percentage of the full space.
>
> As shown in Table R2, although the accumulated raw span can become large, the effective rank remains compact. At the final task, the raw history rank reaches 94.1\%, whereas the effective history rank remains only 14.1\%. This is further supported by the small discarded historical energy, whose maximum is only 0.0032%. Meanwhile, Table R3 shows that the gated gradients consistently retain lower energy on sensitive directions and higher energy on plastic directions than the raw gradients throughout the sequence. This suggests that the gate remains selective rather than degenerating.
>
> **Table R2: Consolidated subspace compactness**
> |Metric|Result(%)|
> |---|---:|
> |Raw rank (Task 40)|94.1|
> |Effective rank (Task 40)|14.1|
> |Max discarded energy (Task 33) |0.0032|
>
> **Table R3: Gating selectivity ($N=40$)**
> |Task|Raw Sensitive(%)|Gated Sensitive(%)|Raw Plastic(%)|Gated Plastic(%)|
> |---|---:|---:|---:|---:|
> |2|46.71|0.00|53.29|100.00|
> |10|81.31|34.48|18.69|65.52|
> |20|87.98|47.78|12.02|52.22|
> |30|92.66|58.65|7.34|41.35|
> |40|94.78|68.30|5.22|31.70|

---

> > ### Author Rebuttal · Reviewer_Uo5j · 2026-04-03
> >
> > I thank the authors for their response. My concerns have been addressed. I am raising my score from 3 to 4.

---

> > > ### Author Response · Authors · 2026-04-03
> > >
> > > Thanks for your acknowledge  of our work and for raising score. Thanks again for your time and effort in reviewing our paper.

---

### Official Review · Reviewer_DnVn · 2026-03-12

**Soundness:** 3
**Presentation:** 3
**Significance:** 3
**Originality:** 3
**Overall Recommendation:** 4
**Confidence:** 4

**Summary:**

MoCL is a rehearsal-free, regularization-based continual learning (CL) approach applied to pre-trained models (PTMs). It employs a novel, metabolic-suppression-inspired gating mechanism that suppresses gradient components along heavy-tailed curvature modes, thereby moving beyond earlier work based on hard gating. It identifies geometric manifolds of consolidated knowledge and constructs Fisher Information Matrix (FIM) subspaces for each task. It proposes a factorized subspace approximation to efficiently estimate the curvature geometry during training. These task-specific subspaces are consolidated over time using Incremental SVD. The paper empirically demonstrates that, through this method, the parameter update trajectory passes through a low-shared-loss region, thereby avoiding catastrophic forgetting. This approach is applied to both linear and nonlinear layers and shown to perform stably in full-parameter training settings. Experimentally, MoCL outperforms previous state-of-the-art methods based on regularization and LoRA in the class-incremental learning setting.

**Compliance With Llm Reviewing Policy:**

Affirmed.

**Final Justification:**

The rebuttal (including new ablation studies, answers to questions, and the promised reframing of some pieces) increased my "soundness" and confidence scores; however, I believe the overall recommendation remains appropriate.

**Key Questions For Authors:**

Q1) In the nonlinear compatibility experiments (Table 8), what precisely in MoCL's formulation makes it compatible with nonlinear layers? Does MoCL apply K-FAC to nonlinear layers without modification, relying on empirical robustness?

Q2) Could the authors report how the rank of the consolidated subspace grows with the number of tasks, during incremental SVD consolidation, and whether any rank truncation strategy is used?

Q3) Does the ablation study include a variant of MoCL with standard K-FAC but without the low-rank momentum augmentation? If not, could the authors provide this?  A positive result (meaningful degradation without low-rank momentum) would strengthen the contribution claim in the first "strength" point above. A null result would reduce the novelty claim but not affect the overall recommendation.

**Limitations:**

The paper did not discuss the limitations of the proposed approach. This is one of the paper's weak points. For example, the lack of a theoretical proof of non-linear compatibility should have been acknowledged. Extension to more complex continual learning scenarios, like Online Continual Learning, could have been discussed, and the limitations of Class Incremental Learning (CIL) should have been acknowledged.

**Strengths And Weaknesses:**

Strengths:
1. Overall, the paper is sound: it proposes a well-motivated curvature approximation method, replacing the diagonal FIM with a K-FAC + low-rank momentum augmentation, a principled upgrade over EWC-style regularization.

2. Soft gating via Tsallis entropy is a novel mechanism. Moving from hard orthogonal projection and gating to a continuous, energy-aware modulation factor is a valuable contribution. It allows for a task-agnostic suppression mechanism that does not require task identity at inference time.

3. Practical extension to nonlinear layers. Many curvature-based CL methods are formally restricted to linear operations or require frozen backbones. MoCL applies its FIM-based gating to full-parameter training, including nonlinear layers, and empirically validates this on ImageNet-R (Table 8), maintaining stability where competitors degrade or run out of memory. The theoretical grounding of this extension remains an open question (see W3), but the practical result is noteworthy.

4. The incremental SVD consolidation merges FIM subspaces across tasks and keeps memory and computation bounded. Unlike previous work, which stores per-task bases independently, causing linear memory growth, MoCL's incremental scheme absorbs new task subspaces into a single evolving low-rank representation, which is a meaningful engineering contribution.

5. Strong empirical results in a class-incremental setting. Outperforming both regularization-based methods and LoRA-based competitors in class-incremental learning on multiple benchmarks is a meaningful empirical result, even if the margin over the best baseline (CoSO) requires better contextualization (see the 1st point in the weaknesses).

6. The paper is well written, the narrative is easy to follow, and it positions itself in well structured context of prior works, while building upon them.


Weaknesses:
1. Oracle and unconstrained fine-tuning accuracies have not been reported to inform the reader about the upper and lower bounds of the performance. Without these it is not clear how significant the 1-3% improvement is compared to the competing method called CoSO.

2. The paper claims to bring efficiency gains; however, in Table 6, compared to CoSO, MoCL brings only ~5% reduction in VRAM, no reduction in GFLOPS, deeming this practically not a significant contribution. The efficiency claim in the abstract should be either removed or substantially qualified.

3. The claim of principled nonlinear compatibility is not theoretically supported. The paper asserts that MoCL's FIM + metabolic gating enables "geometry-aware modulation" for nonlinear layers, but provides no derivation, correction, or theoretical analysis of how K-FAC's Kronecker factorization, which requires linear layer structure and activation-gradient statistical independence, is extended to nonlinear settings. The empirical evidence in Table 8 shows MoCL is more practically robust than competitors, but this does not validate the claim of a principled geometric treatment.

4. No ablation of the low-rank momentum component. The low-rank momentum augmentation to the K-FAC estimate is presented as a novel component, but the paper does not include an ablation isolating its contribution.

These weaknesses are the reasons for the "2 (fair)" rating for the "soundness" criteria. While thanks to the strengths mentioned above, the presentation, significance, and originality rate is "3 (good)".

---

> ### Author Rebuttal · Authors · 2026-03-31
>
> We thank the reviewer for the constructive feedback and respond below.
>
> ## Weaknesses
>
> > **W1:** Missing Oracle and unconstrained fine-tuning results to contextualize the 1–3% gain over CoSO.
>
> **AW1:** We agree that the gain is hard to interpret without references. To address it, we use joint training as an oracle upper bound and full fine-tuning as an unconstrained sequential reference in Table R1. As shown in Table R1, MoCL is consistently closer to joint training than CoSO across all ImageNet-R settings. At $N=20$, the gap in $A_{last}$ to joint training is reduced from 6.01 to 4.36. This shows that the 1-3% gain reduces a substantial part of the remaining oracle gap.
>
> **Table R1: Bound references ($A_{last} / A_{avg}$)**
> |Method|N=5|N=10|N=20|
> |---|---|---|---|
> |Joint|$83.58_{(0.04)}$ / -|$83.58_{(0.04)}$ / -|$83.58_{(0.04)}$ / -|
> |FFT|$62.57_{(0.76)}$ / $76.58_{(0.62)}$|$56.52_{(0.95)}$ / $71.39_{(0.88)}$|$52.67_{(1.17)}$ / $68.46_{(0.82)}$|
> |CoSO|$81.53_{(0.09)}$ / $84.98_{(0.11)}$|$80.33_{(0.16)}$ / $84.78_{(0.18)}$|$77.57_{(0.20)}$ / $82.65_{(0.13)}$|
> |MoCL|$82.29_{(0.11)}$ / $86.32_{(0.09)}$|$81.33_{(0.18)}$ / $86.07_{(0.19)}$|$79.22_{(0.17)}$ / $85.03_{(0.20)}$|
> > **W2:** Efficiency gains are marginal compared to CoSO. Qualify the abstract claim.
>
> **AW2:** We agree and will qualify this claim accordingly. In our results, the practical gain is mainly a modest reduction in VRAM, while GFLOPs remain comparable to CoSO. We will therefore tone down the abstract claim and present efficiency as a limited practical benefit rather than a broad computational advantage.
>
> > **W3 & Q1:** Missing theoretical analysis for MoCL’s nonlinear compatibility and its extension of K-FAC assumptions to non-linear layers.
>
> **AW3 & AQ1:** We do not claim MoCL as an exact K-FAC extension or a complete theory for arbitrary nonlinear operators. Instead, MoCL uses a local second-order view, where the increase in loss on previous tasks is approximated by the FIM and updates are scaled by directional sensitivity rather than hard linear orthogonality.
>
> In the nonlinear setting, MoCL applies the standard layer-wise K-FAC approximation to the trainable linear weight matrices inside the network, rather than to an entire nonlinear module as an exact factorization. For a block
> $s_l=W_la_{l-1}+b_l,\ a_l=\phi_l(s_l)$,
> the gradient with respect to $W_l$ still has the form
> $\nabla_{W_l}\ell=\delta_l a_{l-1}^\top$.
> This gives the usual K-FAC approximation
> $F_l\approx \mathbb{E}[a_{l-1}a_{l-1}^\top]\otimes \mathbb{E}[\delta_l\delta_l^\top]$.
> Thus, even in a nonlinear network, the required linear structure is preserved at the level of each trainable weight matrix, and MoCL uses the resulting approximation for subspace estimation and gating.
>
> That said, activation–gradient independence remains an approximation. MoCL follows the standard K-FAC approximation setting rather than providing an exact theory for arbitrary nonlinear operators.
>
> > **W4 & Q3:** Need an isolated ablation for low-rank momentum.
>
> **AW4 & AQ3:** We ablate low-rank momentum by replacing MLorc with a full-rank optimizer, while keeping the rest of MoCL unchanged. Table R2 shows only small changes across task lengths, so low-rank momentum is not the main source of gain. Instead, the main accuracy gain appears to come from the factorized subspace and metabolic gating.
>
> **Table R2. Momentum ablation ($A_{last} / A_{avg}$)**
> |Method|N=5|N=10|N=20|
> |---|---:|---:|---:|
> |MoCL|82.29 / 86.32|81.33 / 86.07|79.22 / 85.03|
> |w/o low-rank|82.67 / 86.27|82.04 / 86.09|79.32 / 85.24|
> |Delta|+0.38 / -0.05|+0.71 / +0.02|+0.10 / +0.21|
>
> ---
> ## Questions
>
> > **Q2:** How does the subspace rank grow, and what rank truncation strategy is used?
>
> **AQ2:** We use explicit rank truncation during incremental SVD consolidation, where each task subspace is first truncated by the cumulative-energy threshold and rank cap, and the merged historical subspace is then truncated again.
>
> Empirically, the retained effective rank remains compact as tasks accumulate. On ImageNet-R with 40 tasks, the accumulated raw subspace spans 94.1% of the full space, but after consolidation and truncation we retain only 14.1% while discarding at most 0.0032% historical energy, indicating substantial redundancy rather than degeneration to full rank.
>
> ---
> ## Limitations
>
> > **L1:** The limitations are insufficiently discussed, including the lack of non-linear theoretical support and applicability to OCL and broader CIL settings.
>
> **AL1:** Regarding nonlinear compatibility, as noted in W3 & Q1, MoCL has a local justification for nonlinear networks, while a complete theory for arbitrary operators remains future work. More broadly, our current experiments are limited to rehearsal-free CIL. While the second-order formulation may extend beyond CIL, applying MoCL to OCL would require replacing the current task-wise consolidation with a streaming update scheme, which is beyond the scope of this paper. We will clarify these limitations.

---

> > ### Author Rebuttal · Reviewer_DnVn · 2026-04-02
> >
> > I appreciate all the effort that went into answering questions and new experiments, ablation studies. When these are integrated into the manuscript, and contributions are refined and reframed a bit as discussed above, I think the paper will be stronger. These increased my confidence from 3 to 4. I have no further questions.

---

> > > ### Author Response · Authors · 2026-04-03
> > >
> > > Thank you for your positive feedback and for taking the time to review our work. We sincerely appreciate your helpful comments and will refine the paper accordingly.

---

### Official Review · Reviewer_tDRg · 2026-03-13

**Soundness:** 2
**Presentation:** 3
**Significance:** 2
**Originality:** 2
**Overall Recommendation:** 4
**Confidence:** 4

**Summary:**

The paper studies continual learning under catastrophic forgetting, proposing a rehearsal-free framework (MoCL) that combines factorized subspace approximation with Tsallis entropy–based gradient gating to balance stability and plasticity. Overall, the paper presents an incremental improvement over existing subspace-based continual learning methods, but the level of novelty and theoretical justification appears limited.

**Compliance With Llm Reviewing Policy:**

Affirmed.

**Final Justification:**

The authors have thoroughly addressed my concerns, particularly regarding the novelty and the effectiveness of the proposed method. Therefore, I have increased my score.

**Key Questions For Authors:**

1. The paper claims that “MoCL enables the model to converge to a shared low-loss region across sequential tasks.” However, the theoretical analysis mainly relies on local quadratic approximations and upper-bound arguments. It is unclear how these results justify convergence to a shared low-loss region across tasks.
2. The paper adopts Tsallis entropy gating motivated by the heavy-tailed spectrum of the Fisher Information Matrix. However, it remains unclear why Tsallis gating is more appropriate than other continuous spectral weighting functions, rather than simply being an empirically effective choice.
3.Why does a heavy-tailed Fisher Information Matrix (FIM) necessarily imply a better stability–plasticity balance in continual learning?
4. How sensitive are the results to the choice of k?
5. How well does the claim of a “shared low-loss region” generalize? Is it broadly valid, or only observed in specific visualization or interpolation cases?
6. If the goal is to replace orthogonal projection methods, comparisons with OGD/GPM, EWC/NCL, and simple spectral weighting baselines would strengthen the evaluation.

**Limitations:**

The paper briefly discusses computational overhead and approximation assumptions, but the limitations could be elaborated more clearly. In particular, the dependence on curvature approximation, the sensitivity to subspace rank selection, and the scalability to very long task sequences are not sufficiently analyzed.

**Strengths And Weaknesses:**

Soundness: From a technical perspective, the derivations are generally self-consistent. The use of Kronecker-factorized subspace approximation and incremental SVD is computationally reasonable, and the formulation of the gradient gating mechanism is clearly defined. The theoretical analysis relies on local quadratic approximations of the loss and provides upper bounds on the influence of updates on previous tasks, which supports the main intuition of the method.
However, the theoretical results are mainly based on local approximations (e.g., Fisher approximation and low-rank truncation) and therefore serve more as explanatory analysis rather than strict convergence guarantees. In particular, the claim that the model converges to a shared low-loss region across tasks is not theoretically established.
Presentation: The paper is generally well organized, and the overall framework and experimental setup are clear. The figures help illustrate the proposed method. However, the notation in the method section is relatively dense, and some key variables and hyperparameters are not clearly explained. In addition, the descriptions of important steps, such as subspace updating and gating computation, are somewhat brief.
Significance: The paper addresses catastrophic forgetting in continual learning, an important problem in machine learning. However, the proposed method mainly extends existing subspace-based approaches by replacing hard projection with a continuous spectral weighting mechanism. Although the experiments show performance improvements, the overall idea represents a technical refinement of existing methods rather than a substantial advance in the understanding or paradigm of continual learning.
Originality: The method mainly consists of Kronecker-factorized curvature approximation, subspace accumulation and protection, and continuous spectral weighting of gradients. The first two components are closely related to existing approaches such as K-FAC, GPM, and OWM, making the framework largely an extension of existing subspace-based methods. The main change lies in using a K-FAC/FIM-based low-rank subspace instead of first-order constraints and replacing hard orthogonal projection with continuous spectral weighting. Therefore, the overall novelty of the work appears limited.

---

> ### Author Rebuttal · Authors · 2026-03-31
>
> We thank the reviewer for the constructive feedback and respond below.
>
> ## Questions
> > **Q1:** How do the local analysis and upper bounds support the shared low-loss claim?
>
> **AQ1:** Our theoretical analysis is not intended to provide a formal convergence guarantee to a shared low-loss region. Instead, the local analysis and upper bounds explain why MoCL suppresses updates along sensitive directions, which can facilitate shared low-loss behavior.
>
> Thus, the shared low-loss region claim should be interpreted as empirically supported rather than as a formal theorem. In particular, our LMC analysis shows a lower loss barrier along interpolation paths between task solutions. We will revise the Abstract, Introduction, and Secs. 3.2–3.3 to clarify.
>
> > **Q2.1:** Why Tsallis over other continuous spectral gates beyond empirical results?
>
> **AQ2.1:** We do not claim Tsallis is the only possible continuous gate. Rather, our analysis of the forgetting term suggests that a suitable gate should preserve low-curvature directions, suppress dominant spikes, and adapt to the estimated tail of the Fisher spectrum.
>
> In contrast, for $q<1$, the Tsallis gate in Eq. (3) preserves low-curvature modes while imposing a finite cutoff on sharp directions, with its shape linked to the estimated spectrum tail through Eq. (2). Thus, Tsallis is not the only possible gate, but it is a principled choice for this objective because it combines spike suppression and tail adaptation in a single continuous form.
>
> > **Q2.2:** Why does a heavy-tailed FIM imply a better stability–plasticity balance?
>
> **AQ2.2:** Our point is not that a heavy-tailed FIM by itself guarantees a better stability–plasticity balance. Rather, it reveals a geometry with dominant spikes and many sloppy modes, making rigid thresholding suboptimal and motivating a continuous gate informed by the tail of the spectrum. The improved balance comes from using this spectral structure to design the gating mechanism, rather than from heavy-tailedness alone. We will revise the Introduction to make this distinction explicit.
>
> > **Q3:** How sensitive is performance to k?
>
> **AQ3:** In MoCL, $k$ is not manually tuned but selected adaptively from the spectrum knee point. The performance remains highly stable under substantial perturbations of the selected $k$, with $\mathcal{A}\_{\text{last}}$ varying only from 78.72 to 79.25 and $\mathcal{A}\_{\text{avg}}$ from 84.91 to 85.04. Due to the response length limit, the detailed table is provided in our response to Reviewer Uo5j on Q1 (Table R1).
>
> > **Q4:** How general is the shared low-loss region claim beyond specific interpolation cases?
>
> **AQ4:** To further assess the generality, we evaluate multiple adjacent and separated task pairs on ImageNet-R and report the interpolation loss variation in Table R1. MoCL consistently exhibits smaller loss variation than standard Adam across all tested pairs, in both standard deviation and loss range. This suggests that the shared low-loss behavior is not limited to a single interpolation example in our setting.
>
> **Table R1: Interpolation loss variation ($\times 10^{-2}$)**
>
> |Task pair|Std. Adam|Std. MoCL|Range Adam|Range MoCL|
> |---|---:|---:|---:|---:|
> |4-5|2.56|0.37|8.36|1.16|
> |14-15|3.03|0.13|9.58|0.48|
> |0-9|2.99|1.75|10.94|5.05|
> |9-19|3.38|1.58|11.77|5.05|
>
> > **Q5:** Compare with OGD/GPM, EWC/NCL, and spectral weighting baselines to strengthen the evaluation.
>
> **AQ5:** To address this point, we compare MoCL on ImageNet-R ($N=20$) with EWC, GPM, and Sigmoid gating as representative baselines of different families.
>
> As shown in Table R2, MoCL consistently outperforms these baselines, improving $\mathcal{A}\_{\text{last}}$ from 61.82/61.13/74.83 to 79.22 and $\mathcal{A}\_{\text{avg}}$ from 71.54/71.63/82.57 to 85.03. In particular, the Sigmoid comparison suggests that the gain is not solely due to soft weighting, but from MoCL’s curvature-aware gating design.
>
> **Table R2: Baseline comparison**
>
> |Method|$\mathcal{A}_{\text{last}}$|$\mathcal{A}_{\text{avg}}$|
> |:---|:---:|:---:|
> |EWC|$61.82_{(0.74)}$|$71.54_{(0.77)}$|
> |GPM|$61.13_{(0.64)}$|$71.63_{(0.53)}$|
> |Sigmoid|$74.83_{(0.32)}$|$82.57_{(0.29)}$|
> |MoCL|$79.22_{(0.17)}$|$85.03_{(0.20)}$|
> ---
> ## Limitations
>
> > **L1:** Insufficient analysis of curvature dependence, rank sensitivity, and long-sequence scalability.
>
> **AL1:** Regarding curvature dependence, MoCL depends on the fidelity of curvature approximation, while stronger second-order estimators may further improve robustness. Relatedly, it uses energy-based adaptive truncation rather than a fixed manually tuned rank, and performance may still depend on the quality of the adaptive truncation and retained rank budget. For long sequences, our analysis of 40 tasks on ImageNet-R suggests that cumulative degradation remains well controlled in practice. For space reasons, please see our response to Reviewer 9Ap3 on W3 & L1 for details. We will make these limitations more explicit in the revised manuscript.

---

> > ### Author Rebuttal · Reviewer_tDRg · 2026-04-06
> >
> > Thank you to the authors for the detailed response. I will raise my score from 3 to 4.

---

> > > ### Author Response · Authors · 2026-04-06
> > >
> > > We sincerely thank the reviewer for carefully reading our rebuttal and for increasing their score. We greatly appreciate the time and effort you dedicated to reviewing our work and providing constructive feedback. Your acknowledgment has been very encouraging.

---

### Official Review · Reviewer_9Ap3 · 2026-03-13

**Soundness:** 3
**Presentation:** 2
**Significance:** 2
**Originality:** 3
**Overall Recommendation:** 4
**Confidence:** 3

**Summary:**

The paper proposes MoCL (Metabolic Optimization for Continual Learning), a rehearsal-free framework designed to balance stability and plasticity in continual learning. The core innovation lies in a "metabolic gating" mechanism inspired by neurobiology, which utilizes Tsallis entropy to handle the heavy-tailed distribution of the Fisher Information Matrix (FIM). By leveraging a factorized subspace approximation, the method achieves curvature-aware gradient modulation without the prohibitive computational costs typically associated with second-order optimization.

**Compliance With Llm Reviewing Policy:**

Affirmed.

**Key Questions For Authors:**

Please see Strengths And Weaknesses

**Limitations:**

- Although the truncation error is proven to be bounded, the cumulative effect of using factorized approximations over very long task sequences is not fully explored.

**Strengths And Weaknesses:**

Strengths,
- The authors provides a good perspective on gradient modulation in non-linear manifolds, moving beyond simple linear orthogonal projections.
- The use of factorized subspace approximation and incremental SVD allows the model to capture essential curvature information with minimal memory overhead.
- MoCL consistently outperforms state-of-the-art methods (e.g., InfLoRA, COSO) across various benchmarks.

Weaknesses,
- While the gating mechanism modulates updates based on curvature, the temporal evolution of gradient flows across tasks remains less clear.
- The metabolic temperature $\tau$ and the entropy index $q$ appear to be critical to performance.
- Although the truncation error is proven to be bounded, the cumulative effect of using factorized approximations over very long task sequences is not fully explored.
- The review of related work on continual learning should be strengthened, such as focusing on C-Flat [NeurIPS 24], C-Flat Turbo [CVPR 26] and ZeroFlow [ICML 25] in optimization, which deal with geometry and gradients.

---

> ### Author Rebuttal · Authors · 2026-03-31
>
> ## Weaknesses and Limitations
> > **W1:** While the gating mechanism modulates updates based on curvature, the temporal evolution of gradient flows across tasks remains less clear.
>
> **AW1:** We appreciate the reviewer’s suggestion to clarify the temporal evolution of gradient flows. To address this, we analyze the projected composition of raw and gated gradients in the historical subspace, and track how the sensitive and plastic energy ratios evolve as tasks accumulate.
>
> As shown in Table R1, the gated gradients consistently exhibit lower energy ratios along historical sensitive directions than the raw gradients, while retaining higher energy ratios along plastic directions as tasks accumulate. Although the gap narrows as tasks accumulate, it remains clearly positive even in later tasks, indicating that the gate remains selective throughout the sequence. This suggests that MoCL selectively suppresses gradient flow along historical sensitive directions while preserving more plastic directions throughout the sequence.
>
> **Table R1: Evolution of gradient flows**
>
> |Task|Raw Sensitive (%)|Gated Sensitive (%)|Raw Plastic (%)|Gated Plastic (%)|
> |---|---:|---:|---:|---:|
> |2|45.62|0.0|54.38|100.0|
> |4|67.88|13.15|32.12|86.85|
> |6|81.52|37.90|18.48|62.10|
> |8|91.31|70.31|8.69|29.69|
> |10|93.56|73.85|6.44|26.15|
>
> > **W2:** The metabolic temperature $\tau$ and the entropy index $q$ appear to be critical to performance.
>
> **AW2:** We appreciate the reviewer’s observation that $\tau$ and $q$ are important to performance. However, our results suggest that MoCL is not overly sensitive to their settings in practice.
>
> For $\tau$, MoCL remains stable under moderate variations rather than relying on a narrow choice. As shown in Fig. 5 in the appendix, the performance stays high from $\tau=10^{-6}$ to $\tau=10^{-3}$, and only degrades clearly when $\tau$ becomes larger than $10^{-2}$, where the gate becomes overly smooth and gradually approaches unconstrained fine-tuning.
>
> For $q$, MoCL does not treat it as an independently tuned hyperparameter. Instead, it is induced from the estimated tail structure, where $k$ determines how many leading singular values are used for tail estimation and is selected adaptively from the spectrum knee point. We therefore assess the robustness of the induced $q$ by varying the automatically selected $k$ on ImageNet-R ($N=20$) from $0.5k$ to $1.5k$. As shown in Table R2, performance remains highly stable, with $\mathcal{A}\_{\text{last}}$ varying only from 78.72 to 79.25 and $\mathcal{A}\_{\text{avg}}$ from 84.91 to 85.04, indicating that MoCL is robust to moderate changes in the estimation range and that the induced $q$ remains stable under such perturbations.
>
> **Table R2: Sensitivity Analysis of $k$, where varying $k$ induces corresponding changes in $q$**
>
> |Metric(%)|$0.5k$|$0.75k$|$0.9k$|$1.0k$|$1.1k$|$1.25k$|$1.5k$|
> |:---|:---:|:---:|:---:|:---:|:---:|:---:|:---:|
> |$\mathcal{A}_{\text{last}}$|$78.93_{(0.11)}$|$78.98_{(0.13)}$|$78.72_{(0.22)}$|$79.22_{(0.17)}$|$79.25_{(0.17)}$|$78.93_{(0.16)}$|$79.07_{(0.14)}$|
> |$\mathcal{A}_{\text{avg}}$|$84.95_{(0.15)}$|$84.92_{(0.16)}$|$84.95_{(0.20)}$|$85.03_{(0.20)}$|$85.04_{(0.21)}$|$84.91_{(0.18)}$|$84.95_{(0.18)}$|
>
> > **W3 & L1:** The cumulative effect of using factorized approximations over very long task sequences is not fully explored.
>
> **AW3 & AL1:** We appreciate the reviewer’s concern regarding the cumulative effect. We therefore directly track the historical energy discarded by repeated capped merging, which quantifies the additional fraction of historical spectral energy removed when the accumulated historical subspace is repeatedly recompressed under the rank cap.
>
> As summarized in Table R3, this quantity remains very small throughout 40 tasks on ImageNet-R, with a maximum of only 0.0032% at Task 33. This suggests that repeated factorized consolidation remains well controlled in practice and does not exhibit substantial cumulative degradation over long task horizons.
>
> **Table R3: Historical energy discarded**
> |Task|Discarded|
> |---|---:|
> |Tasks 1-11|0.00% ~ 0.0015%|
> |Tasks 12-40|0.0007% ~ 0.0032%|
> |Maximum at Task 33|0.0032%|
>
> > **W4:** Strengthen literature review on CL optimization, such as C-Flat, C-Flat Turbo and ZeroFlow.
>
> **AW4:** We thank the reviewer for pointing out these relevant works. These works provide a complementary perspective for understanding forgetting and stability.
>
> Specifically, C-Flat and C-Flat Turbo improve continual adaptation by encouraging flatter optima, while ZeroFlow mitigates forgetting in settings where gradient information is unavailable. In contrast, MoCL uses historical second-order curvature to identify directions sensitive to previous tasks and selectively gates updates during sequential learning.
>
> We will make this distinction clearer in the related work discussion of the revised manuscript

---

> > ### Author Rebuttal · Reviewer_9Ap3 · 2026-04-02
> >
> > I appreciate the additional experiments and the thorough efforts made in the rebuttal. Hope that my comments and suggestions will be helpful in further improving the manuscript. I have no further questions.

---

> > > ### Author Response · Authors · 2026-04-03
> > >
> > > Thank you for your positive feedback. We sincerely appreciate your time and thoughtful review, and we are glad that our responses have addressed your concerns.

---

### Decision · Program_Chairs · 2026-04-30

**Decision:**

Accept (regular)

**Comment:**

I recommend Weak Acceptance.

Reviewers found the paper technically sound and supported by convincing empirical results. They agreed that the paper addresses a meaningful limitation of hard projection methods in rehearsal-free continual learning, and that the proposed curvature-aware alternative is supported by solid results across standard benchmarks. The rebuttal addressed most of the earlier concerns, and in particular helped clarify the method’s motivation, novelty, and empirical behavior.

The remaining concerns are limited. Reviewer Uo5j suggested a clearer discussion of the relationship to SGP. The authors are encouraged to incorporate the main clarifications from the rebuttal into the paper